# Inflammation and immune activation are associated with risk of *Mycobacterium tuberculosis* infection in BCG-vaccinated infants

Iman Satti[1], Rachel E. Wittenberg [1,6], Shuailin Li[1,6], Stephanie A. Harris[1], Rachel Tanner[1], Deniz Cizmeci [1], Ashley Jacobs[1,2], Nicola Williams [3], Humphrey Mulenga[4], Helen A. Fletcher [5], Thomas J. Scriba [4], Michele Tameris [4], Mark Hatherill [4] & Helen McShane [1] ✉

Tuberculosis vaccine development is hindered by the lack of validated immune correlates of protection. Exploring immune correlates of risk of disease and/or infection in prospective samples can inform this field. We investigate whether previously identified immune correlates of risk of TB disease also associate with increased risk of *M.tb* infection in BCG-vaccinated South African infants, who became infected with *M.tb* during 2-3 years of follow-up. *M.tb* infection is defined by conversion to positive reactivity in the Quanti-FERON test. We demonstrate that inflammation and immune activation are associated with risk of *M.tb* infection. Ag85A-specific IgG is elevated in infants that were subsequently infected with *M.tb*, and this is coupled with upregulated gene expression of immunoglobulin-associated genes and type-I interferon. Plasma levels of IFN-α2, TNF-α, CXCL10 (IP-10) and complement C2 are also higher in infants that were subsequently infected with *M.tb*.

Tuberculosis (TB), caused by infection with *Mycobacterium tuberculosis* (*M.tb*), is a global health threat that, until the COVID-19 pandemic, killed more people annually than any other infectious pathogen. In 2020, 9.9 million new cases and 1.5 million deaths were due to TB[1]. In most *M.tb* infected individuals, infection is either eliminated or contained by the host immune response resulting in a status of asymptomatic infection[2,3]. Bacille Calmette-Guérin (BCG), the only licenced vaccine against TB, does not adequately protect against pulmonary disease, with especially low efficacy in TB endemic areas; hence, a new and improved vaccine is urgently needed[4].

Development of an effective and efficacious TB vaccine would be facilitated by an improved understanding of protective immunity. Although many aspects of host immunity are known to play a role in protection against TB, defined and validated correlates of protection are lacking. The identification of immune correlates of protection in human studies would allow for the rational design and development of TB vaccine candidates, and antigen delivery systems could be optimised to induce these immune correlates. Validation of immune correlates in field efficacy trials would ultimately enable the development and licensure of vaccines without the need for expensive, time consuming field efficacy studies.

[1]Jenner Institute, Nuffield Department of Medicine, University of Oxford, Oxford OX3 7DQ, UK. [2]Wellcome Centre for Infectious Diseases Research in Africa, University of Cape Town, Observatory, Cape Town, South Africa. [3]Nuffield Department of Primary Care Health Sciences, University of Oxford, Radcliffe Observatory Quarter, Woodstock Road, Oxford OX2 6GG, UK. [4]South African Tuberculosis Vaccine Initiative (SATVI), Institute of Infectious Disease and Molecular Medicine and Division of Immunology, Department of Pathology, University of Cape Town, Cape Town, South Africa. [5]Department of Infection Biology, Faculty of Infectious and Tropical Diseases, London School of Hygiene & Tropical Medicine, London, UK. [6]These authors contributed equally: Rachel E. Wittenberg, Shuailin Li. ✉e-mail: helen.mcshane@ndm.ox.ac.uk

In the absence of sample sets from trials of effective TB vaccines, where correlates of protection can be identified, exploring immune correlates of risk in prospective samples can inform this field. Although MVA85A did not confer improved efficacy over BCG vaccination alone, samples collected from BCG-vaccinated South African infants who participated in the MVA85A efficacy trial provide a good opportunity for studying correlates of risk of TB disease and *M.tb* infection in this important target population[5]. We have previously shown, in a case-control study investigating immune correlates of risk of TB disease in the same cohort of South African infants, that lower BCG-specific ex-vivo IFN-γ responses, lower Ag85A-specific IgG antibodies and higher non-antigen specific CD4[+] T-cell activation were all associated with an increased risk of TB disease[6]. We have identified transcriptomic signatures that correlated with progression from *M.tb* infection to TB disease in South African adolescents[7]. Measurement of these signatures has been translated to microfluidic RT-qPCR, allowing more efficient application to further studies[7]. Persistent viral or bacterial infections can result in T-cell activation and dysfunction of antigen-specific T-cells[8–10]. We have previously studied the association between human cytomegalovirus (CMV) infection and T-cell activation using samples from the same cohort of South African infants. We found that a CMV-specific IFN-γ response was associated with CD8[+] T-cell activation and with increased risk of subsequent TB disease[11]. Similarly, CMV antibody responses were associated with increased risk of TB in Ugandan adults[12]. CMV infection acquired in the first year of life is associated with increased risk of TB disease in South African infants[13]. Respiratory viral infections have also been shown to be associated with increased risk of TB progression and to drive conversion to a positive TB disease gene signature[14].

Given the time and expense required to evaluate candidate TB vaccines in Prevention of Disease (PoD) efficacy trials, there has been a recent shift of focus to Prevention of *M.tb* Infection (PoI) studies, as a way of evaluating the efficacy of candidate TB vaccines in humans prior to PoD efficacy trials[15]. As only ~5-10% of infected individuals progress to active disease during their lifetime, the incidence of *M.tb* infection is higher than the incidence of TB disease. PoI studies therefore have more endpoints and are thus shorter and more cost-effective than PoD studies[16]. The PoI approach was utilised in an efficacy trial in which BCG revaccination conferred protection of South African adolescents against sustained *M.tb* infection[17]. A possible assumption implicit in this approach is that the same immunological mechanisms confer protection against both TB disease and *M.tb* infection. It is important to determine the validity of this assumption.

In our current study, we examined samples collected from South African infants who participated in the MVA85A efficacy trial[5] to determine whether the correlates of risk of TB disease we identified in this population[6] are also correlates of risk of *M.tb* infection. We characterised baseline immune responses, prior to acquisition of *M.tb* infection, in two groups of BCG-vaccinated 4-6 months old infants: infants who went on to become infected with *M.tb* during a follow-up period of 2-3 years, but who did not develop disease, and infants who remained uninfected throughout the study, referred to as *M.tb*-infected and *M.tb*-uninfected, respectively. *M.tb* infection was defined using the QuantiFERON (QFT) TB Gold In-tube test, which has been used extensively in TB endemic and non-endemic countries to define *M.tb* infection[18].

In this study we show that factors associated with increased risk of TB disease in South African infants[6] are not associated with increased risk of *M.tb* infection in this cohort. However, we identified inflammation and activation immune markers as well as gene transcripts that are associated with susceptibility to *M.tb* infection in South African infants.

## Results

### QuantiFERON-TB Gold In-Tube conversion

*M.tb* infection was assessed by measuring mycobacteria-specific IFN-γ production using the QFT test, performed prior to enrolment, at Day 336 and at the end of study. All infants had a negative QFT response at enrolment. 15/43 had a positive QFT ≥ 0.35 IU/ml on Day 336 but only two of these remained QFT positive at end of study; they had QFT values of 2.54 IU/ml and 0.82 IU/ml on Day 336 and 5.05 IU/ml and 1.33 IU/ml at end of study, respectively. 20/43 infants who were QFT negative at Day 336 had QFT values ≥ 0.35 IU/ml at the end of study. 5/43 infants converted to positive QFT at time points other than Day 336 and end of study (Fig. 1).

### Ag85A-specific IgG response on Day −7 is associated with increased risk of *M.tb* infection

Baseline characteristics used to investigate possible associations between the studied immune parameters and *M.tb* infection are presented for *M.tb*-infected and uninfected infants (Supplementary Data 1). The number analysed was dependent on the variables used in the model and how many samples were missing for each of the included variables. The primary aim of this study was to investigate if the three immune correlates of risk of TB disease identified in our previous study in the same population were similarly associated with

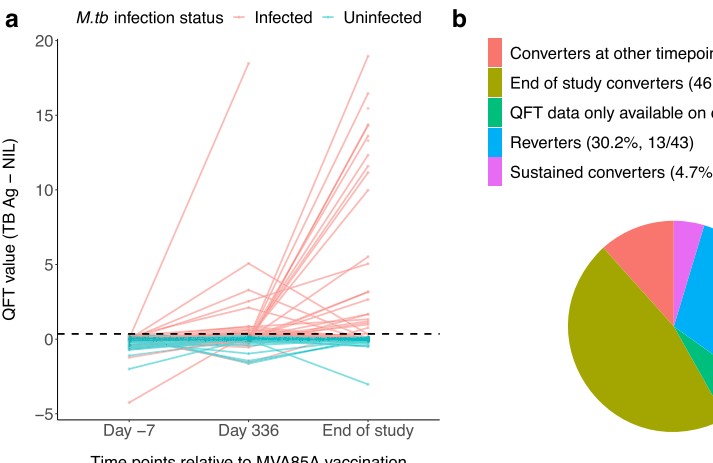

**Fig. 1 | QuantiFERON-TB Gold In-Tube conversion of infants. a** QFT over time for all *M.tb*-infected infants with quantitative QFT, QFT IFN-γ is presented in IU/ml on Day −7, Day 336 and end of study. Each line represents an infant. **b** Conversion and reversion data for all *M.tb*-infected shown as percentages of total *M.tb* infected infants. Source data are provided as a Source Data file.

risk of *M.tb* infection[6]. We evaluated HLA-DR expression on CD4+ T-cells, BCG-specific IFN-$\gamma$ producing cells in PBMC and plasma Ag85A-specific IgG responses. HLA-DR expression on CD4+ T-cells (OR 1, 95% CI 0.95 – 1.06, $p = 0.921$, FDR = 0.921, adjusted only for the three immune correlates of TB disease, same for the remaining two parameters) and BCG-specific IFN-$\gamma$- producing cells (OR 0.99, 95% CI 0.97 – 1.01, $p = 0.238$, FDR = 0.357) were not associated with risk of *M.tb* infection, whereas Ag85A-specific IgG response on Day −7 was associated with increased risk of *M.tb* infection (OR 8.85, 95% CI 2.05 – 38.09, $p = 0.003$, FDR = 0.009).

### Systematic upregulation of cytokines, chemokines and complement component factors in infants with subsequent *M.tb* infection

As an exploratory analysis, we quantified a number of cytokines, chemokines, and complement factors in plasma samples collected from the study subjects in a multiplex assay. The statistical significance of these immune parameters was much higher than those measured by other assays including Enzyme-Linked ImmunoSpot (ELISpot), flow cytometry and Mycobacteria Growth Inhibition Assay (MGIA) (Fig. 2a, Supplementary Data 2). Almost all of the immune parameters measured by the multiplex assay had an odds ratio >1, indicating that the

increased values of these immune parameters were associated with an increased risk of *M.tb* infection (Fig. 2a, Supplementary Data 2). Given that the immune parameters measured by the multiplex assay were closely correlated with each other (Fig. 2b), it is unsurprising that almost all of them have a consistent odds ratio. In addition, the AUROC values of the immune parameters measured by the multiplex assay were also higher than those measured by other assays (Fig. 2c, Supplementary Data 2). Therefore, it is likely that multiple testing correction with other immune parameters caused the loss of statistical power of the immune parameters measured by the multiplex assay. On the basis of these observations, we performed multiple testing correction within the immune parameters measured by the multiplex assay. The results showed that 26 out of the 45 available immune parameters measured by the multiplex assay had an FDR of no greater than 0.2, indicating that systematic upregulation of chemokines, cytokines and complement components in the plasma was associated with the increased risk of *M.tb* infection (Supplementary Data 3).

For all immune parameters except for Ag85A-specific IgG and *M.tb* H37Rv whole cell lysate-specific IgG responses (for which a more comprehensive sample set of plasma was available), the majority of PBMC samples were from Day −7 except when they were not available. All samples used to measure these immune parameters in *M.tb*-

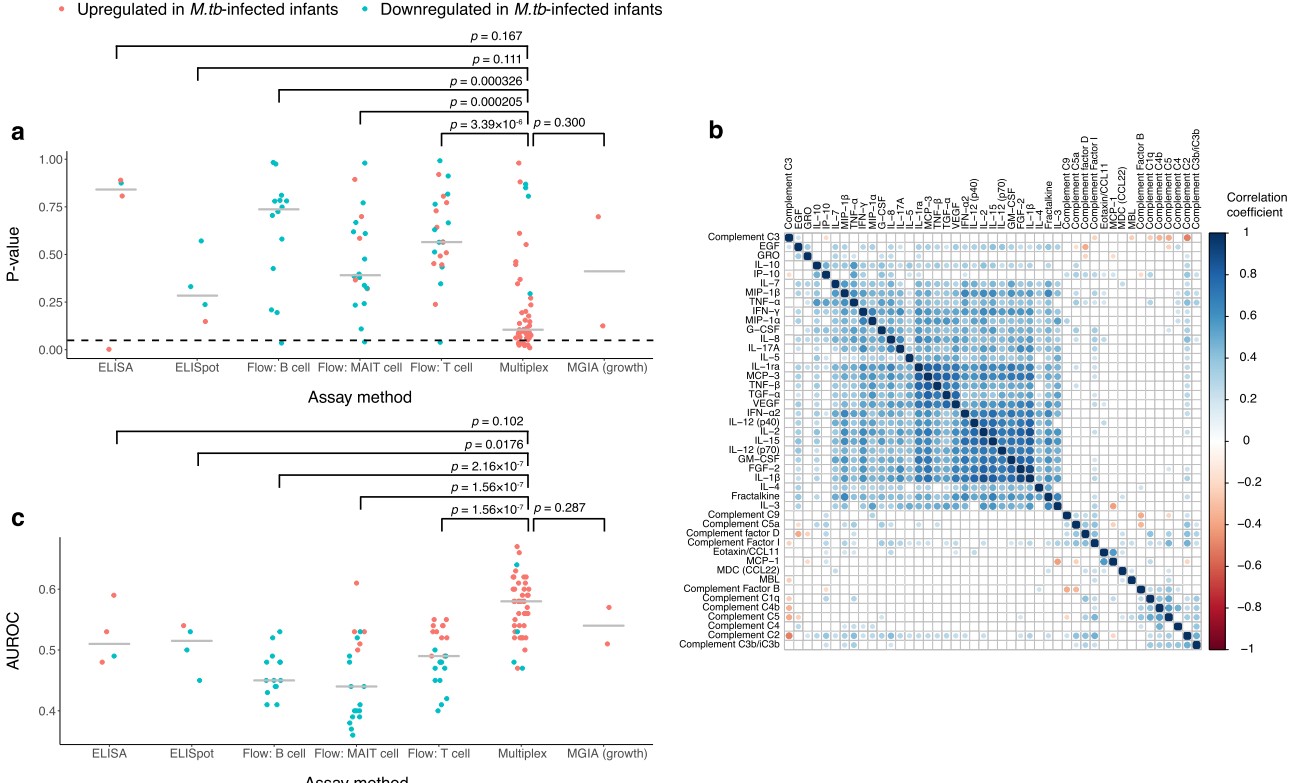

**Fig. 2 | Univariate conditional logistic regression results. a** Unadjusted *p* values of immune parameters measured by different methods. Each point represents an immune parameter. Different columns include immune parameters measured by different assays. Immune parameters included in each column are listed in Supplementary Data 2. The red and blue points represent immune parameters upregulated and downregulated in *M.tb*-infected infants, respectively. The grey bar represents the median value of *p* values of immune parameters measured by the corresponding assay, *n* = 4, 4, 14, 20, 23, 47 and 2 independent immune parameters for Enzyme-Linked ImmunoSorbent Assay (ELISA), ELISpot, Flow: B cell, Flow: Mucosal-Associated Invariant T (MAIT) cell, Flow: T cell, Multiplex and MGIA (growth), respectively. The dotted line represents the *p* value threshold of 0.05. The horizontal axis shows unadjusted *p* values of difference between *M.tb*-infected and *M.tb*-uninfected infants for different immune parameters. Overlay unadjusted *p* values obtained by two-sided Mann-Whitney test show the statistical significance of

difference between the unadjusted *p* values of immune parameters measured by different assays. **b** Spearman correlation between plasma levels of cytokines, chemokines and complements. The colour intensity indicates the correlation coefficient. Only significant correlations are shown in the figure. *P* values were adjusted using Benjamini-Hochberg multiple testing correction. Hypothesis testing used to generate *p* values for the spearman correlation is two-sided. **c** AUROC values of immune parameters measured by different assays. The higher the AUROC value, the better the immune parameter can classify *M.tb*-infected and *M.tb*-uninfected infants. The grey bar represents the median value of AUROC values of immune parameters measured by the corresponding assay, *n* = 4, 4, 14, 20, 23, 47 and 2 independent immune parameters for ELISA, ELISpot, Flow: B cell, Flow: MAIT cell, Flow: T cell, Multiplex and MGIA (growth), respectively. Two-sided Mann-Whitney test without multiple testing correction was done to compare AUROC values of immune parameters measured by different assays. Source data are provided as a Source Data file.

uninfected infants were collected on Day −7, whereas one third of the blood samples from infants who went on to be infected with *M.tb* were collected on Day 28 after MVA85A or placebo vaccination. To investigate the possible confounding effect of the sample collection time points on the univariate conditional logistic regression, we compared results from Day −7 samples only with those from combined Day −7 and Day 28 samples from infants who went on to be infected with *M.tb*. We selected parameters that are different between Day 28 and Day −7 samples from infants who would be infected with *M.tb* (unadjusted *p* values less than 0.1, Supplementary Data 4a), which consisted of a small proportion of all tested immune parameters (21 of 108). The odds ratios, calculated by univariate conditional logistic regression, of most (13 of 21) of these parameters using Day −7 samples only did not change qualitatively (namely from less than 1 to larger than 1 or vice versa) compared to those using all samples (combined Day −7 and Day 28) (Supplementary Fig. 1a and Supplementary Data 4b). Of the immune parameters which were significantly different between *M.tb*-infected and *M.tb*-uninfected infants when we analyzed Day 28 and Day −7 data combined (Supplementary Data 3), only Complement Factor I and Complement C5 lost significance when we looked at data from Day −7 only (Supplementary Fig. 1b and Supplementary Data 4b). Of the immune parameters which were not significantly different between *M.tb*-infected and *M.tb*-uninfected infants (Supplementary Data 2), none showed significance when we only included the Day −7 data. Taken together, the results suggested that most of our analysis of immune correlates of risk of infection were not confounded by the time point of sample collection.

For Ag85A-specific IgG (MVA85A-vaccinated group: OR 1.62, 95% CI 0.69 − 3.77, unadjusted *p* value = 0.261. Placebo group: OR 0.64, 95% CI 0.26 − 1.60, unadjusted *p* value = 0.342) and *M.tb* H37Rv whole cell lysate-specific IgG (MVA85A-vaccinated group: OR 1.28, 95% CI 0.58 − 2.80, unadjusted *p*-value = 0.542. Placebo group: OR 1.17, 95% CI 0.63 − 2.18, unadjusted *p*-value = 0.613) responses on Day 28, we analysed the MVA85A-vaccinated group and placebo group separately, but none of the results were significant, suggesting that they were not confounded by MVA85A vaccination.

## CMV infection is not associated with increased risk of *M.tb* infection

Cellular and humoral immune responses to CMV were measured by IFN-γ ELISpot and IgG and IgM ELISA, respectively. We used the manufacturer's recommendation to determine the cut-off for a positive CMV-specific IgM response. For the CMV-specific IgG response, considering that IgG antibodies can be vertically transmitted from mothers to babies[19], we introduced a more stringent cut-off criteria: only infants who were IgG positive (according to the manufacturer's cut-off) on Day 28 and maintained ≥ 90% of their Day −7 IgG responses on Day 28 were considered CMV-specific IgG positive (Fig. 3a). Infants who had a CMV-specific IFN-γ response of >17 SFC/$1 \times 10^6$ PBMC were considered positive, based on the cut-off used in our previous study on samples collected from the same population[11] (Fig. 3b). All infants who had a positive CMV-specific IgM response and all with positive CMV-specific IFN-γ responses, except one infant who maintained 85% rather than 90% of their Day −7 CMV-IgG response on Day 28, had a positive CMV-specific IgG response, indicating that measuring IgG is more sensitive in detection of CMV infection (Fig. 3c). Therefore, we defined the CMV infection status of infants as follows: infants were defined as CMV-uninfected if they had a negative CMV-specific IgG response; infants were defined as CMV-infected if they had either a positive IgG response, or IgM response or IFN-γ response; if the infant had negative IgM and IFN-γ responses with an unknown IgG response because of sample unavailability, they were considered 'unknown' (Fig. 3c). Using this definition, *M.tb* infection incidence (based on QFT conversion) was 33.9% (21/62) in the CMV-infected group, compared to 16.7% (6/36) in the CMV-uninfected group (*p* = 0.0699, Conditional logistic regression

(OR 2.607, 95% CI 0.9252-7.344), not corrected for multiple testing) (Fig. 3d).

Although we did not detect a direct association between the 3 immune correlates of risk of TB disease and risk of *M.tb* infection, we detected increased frequencies of HLA-DR$^+$ CD4$^+$ T-cells and HLA-DR$^+$ CD8$^+$ T-cells and decreased levels of BCG-specific IFN-γ-producing cells in CMV-infected infants (Supplementary Fig. 2 and Supplementary Data 5).

To ensure that the increase of immune parameters measured by multiplex assays in *M.tb*-infected infants compared to *M.tb*-uninfected infants was not confounded by the CMV infection status of the infants, we evaluated the association between these immune parameters and CMV infection and found that IP-10, TNF-α and complement C2 were all significantly upregulated in CMV-infected infants, compared with CMV-uninfected infants (FDR ≤ 0.2, adjusted only within immune parameters measured by multiplex assays; Supplementary Data 5b). The plasma levels of these three proteins were also significantly upregulated in infants who became infected with *M.tb* (FDR ≤ 0.2, Supplementary Data 3). Only 3 out of the 45 immune parameters were significantly associated with CMV infection, compared to 26 out of the 45 immune parameters that were significantly associated with risk of subsequent *M.tb*-infection, suggesting that the association between most of the immune parameters measured by multiplex assays and risk of *M.tb*-infection was not confounded by CMV infection.

## Differential gene expression between infants with subsequent *M.tb* infection and uninfected infants

Gene expression analysis was performed in samples from 21 *M.tb*-infected infants who had QFT values ≥ 0.35 IU/ml and 52 matched *M.tb*-uninfected controls. Day 28 samples were used for 2 out of the 21 *M.tb*-infected because of lack of samples. When cell availability allowed, each infant was represented by a set of 2 samples: BCG-stimulated and unstimulated. A linear model was fitted to determine differential gene expression using the matched case-control set number as the strata variable and adjusted for the stimulus (BCG stimulated or unstimulated), given that BCG stimulation had a strong influence on gene expression. After differential gene expression analysis (Supplementary Fig. 3a, Supplementary Data 6a), we performed gene set enrichment analysis using two methods and gene set annotation. The first method was the CERNO algorithm using gene sets adapted from Li et al.[20]. The second method was overrepresentation analysis using gene sets defined in the Gene Ontology (GO) project.

The number of infants who became infected with *M.tb* was higher in the CMV-infected group, though this did not reach statistical significance. We therefore performed differential gene expression analysis in CMV-infected and CMV-uninfected infants separately (Supplementary Fig. 3b, c, Supplementary Data 6b, c). We also identified transcripts differentially expressed between CMV-infected and CMV-uninfected infants (Supplementary Fig. 3d, Supplementary Data 6d).

When we analysed combined data from CMV-infected and CMV-uninfected infants, gene sets associated with B-cells, immunoglobulins, complement activation, type I interferon response and anti-viral response were upregulated in infants who were subsequently infected with *M.tb*. These included genes such as *IGKV4-1*, *TNFRSF17*, *IFIH1* and *IFIT2* (Fig. 4a, Supplementary Fig. 4a, Supplementary Data 6a, 7, 8).

In CMV-infected infants, gene sets associated with anti-viral response, type-I interferon response, B-cells, immunoglobulins, complement activation, neutrophil activation and cytokine production and secretion were upregulated in infants who were subsequently infected with *M.tb* compared with those who remained *M.tb*-uninfected, these included genes such as *DDX58*, *IFIH1*, *IFI35*, *CD19*, *IGKV4-1*, *C5*, *C3AR1*, *FUCA2*, *NCF4* and *IL-10* (Fig. 4a, Supplementary Fig. 4b, Supplementary Data 6b, 7, 8).

In CMV-uninfected infants, gene sets related to complement activation, B-cells and immunoglobulins were upregulated in infants

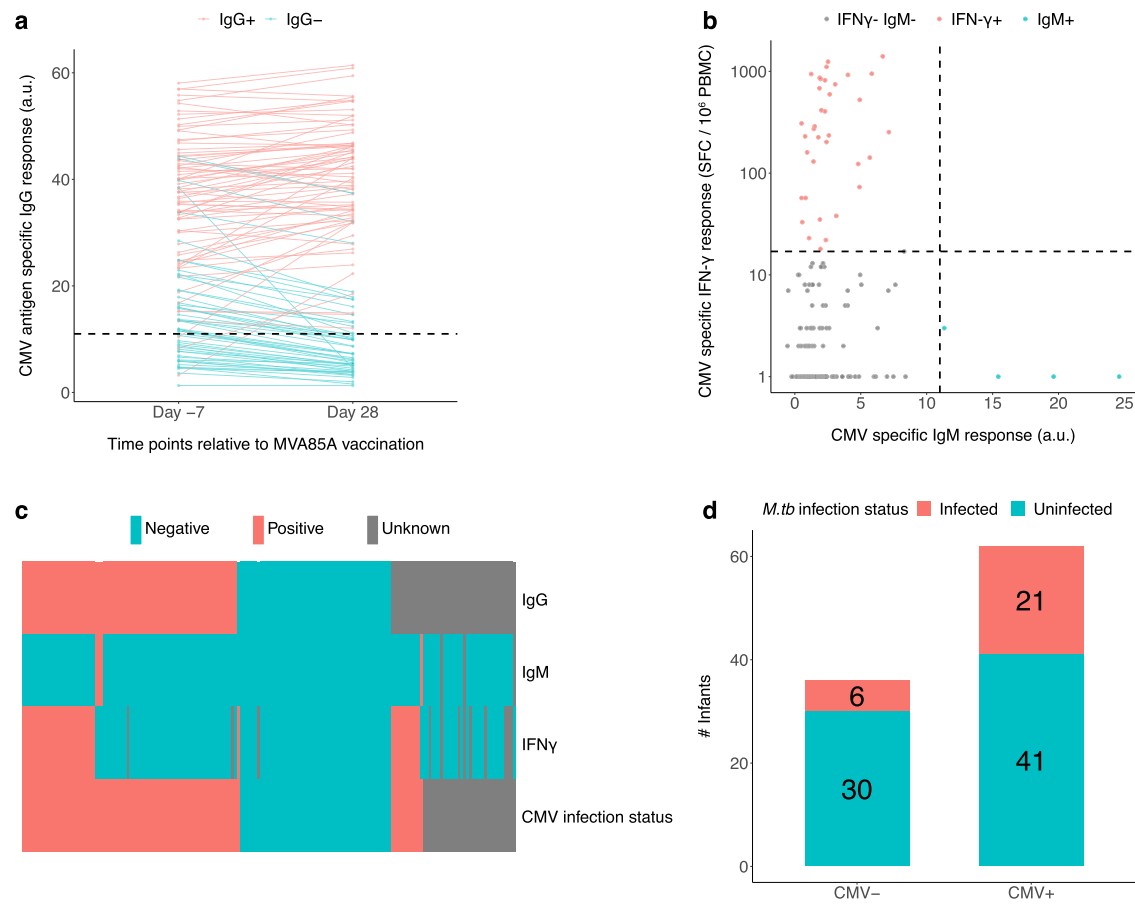

**Fig. 3 | Cytomegalovirus (CMV) reactivity and rate of *M.tb* infection in CMV-reactive (CMV-infected) and non-reactive (CMV-uninfected) BCG-vaccinated South African infants. a** CMV-specific IgG response measured on Day −7 and Day 28 of the study. The horizonal line represents the kit manufacturer's recommended CMV-specific IgG positivity threshold; all infants with CMV-specific IgG below this value on day 28 were considered to have negative IgG reactivity to CMV (IgG-). Above this line, only infants who maintained 90% of their Day −7 CMV-IgG levels on Day 28 were considered to be IgG positive. *N* = 127 biologically independent samples for each time points. **b** CMV-specific IgM and IFN-γ response: The vertical line represents the kit manufacturer's recommended CMV-specific IgM positivity threshold. The horizontal line represents the threshold of CMV-specific IFN-γ response, which is 17 SFC/million PBMC. **c** CMV antigen-specific IgG, IgM and IFN-γ responses are shown in the first 3 rows. The last row is the CMV infection status of infants defined as detailed in the Results section. Red: positive, blue: negative and grey: unknown. **d** Rate of *M.tb* infection in CMV-infected and CMV-uninfected BCG-vaccinated South African infants. Source data are provided as a Source Data file.

that were subsequently infected with *M.tb*, these included genes such as *C1QBP*, *CR2*, *C4BPB*, *CD19* and *IGKV4-1* (Fig. 4a, Supplementary Fig. 4c, Supplementary Data 6c, 7, 8). Gene sets related to neutrophil activation, natural killer cell activation, complement activation were downregulated in infants who were subsequently infected with *M.tb*, these included genes, such as *C1QA*, *C1QB*, *C1QC*, *C3*, *C3AR1*, *C5AR1*, *C5AR2*, *FUCA2*, *KLRB1*, *KLRD1*, *KLRC3* and *NKG7* (Fig. 4a, Supplementary Fig. 4d, Supplementary Data 6c, 7, 8).

Many genes that were differentially expressed between *M.tb*-infected infants and *M.tb*-uninfected infants in CMV-infected infants had the opposite patterns of fold change between *M.tb*-infected and *M.tb*-uninfected infants in CMV-uninfected infants (Fig. 4b). This led to a lower number of differentially expressed genes, when we analysed combined data from CMV-infected and CMV-uninfected infants, compared to when we analysed them separately by CMV infection status. Gene sets associated with *M.tb* infection were strikingly different among CMV-infected and CMV-uninfected infants (Fig. 4a and Supplementary Fig. 4b–d).

To further validate our RNA-seq results, we analysed a dataset from the training set of *M.tb*-infected adolescents in the South African Adolescent Cohort Study (ACS), who either progressed to active TB or remained disease-free[7]. Because the gene expression abundance in the ACS was quantified at the level of splice junction, we used the sum of all splice junction counts of each gene as the read count of the gene and performed differential gene expression analysis using DESeq2. Considering the high prevalence of CMV infection in South Africa[21], we compared the results in the ACS and our CMV-infected infants. We found that gene sets related to complement activation, neutrophils, anti-viral response and type-I interferon response were also upregulated in TB disease progressors in the ACS, which included genes such as *CR1*, *C1QA*, *C1QB*, *C1QC*, *C2*, *C5AR1*, *NCF4*, *IFIH1*, *DDX58*. However, gene sets related to B-cells and immunoglobulins were downregulated in progressors in ACS, though most of the differentially expressed genes in these sets were not immunoglobulin genes, which differs from those observed in our study (Fig. 4c and Supplementary Data 9).

To investigate the association between CMV infection and susceptibility to *M.tb* infection we looked at the overlap between gene sets that were enriched in the differential expression analysis between *M.tb*-infected and *M.tb*-uninfected infants and also between CMV-infected and CMV-uninfected infants. The results showed overlap of differential expression of genes that were associated with B-cells and immunoglobulins (Fig. 4a and Supplementary Fig. 4a, 4e, f, Supplementary Data 6a, d, 7, 8).

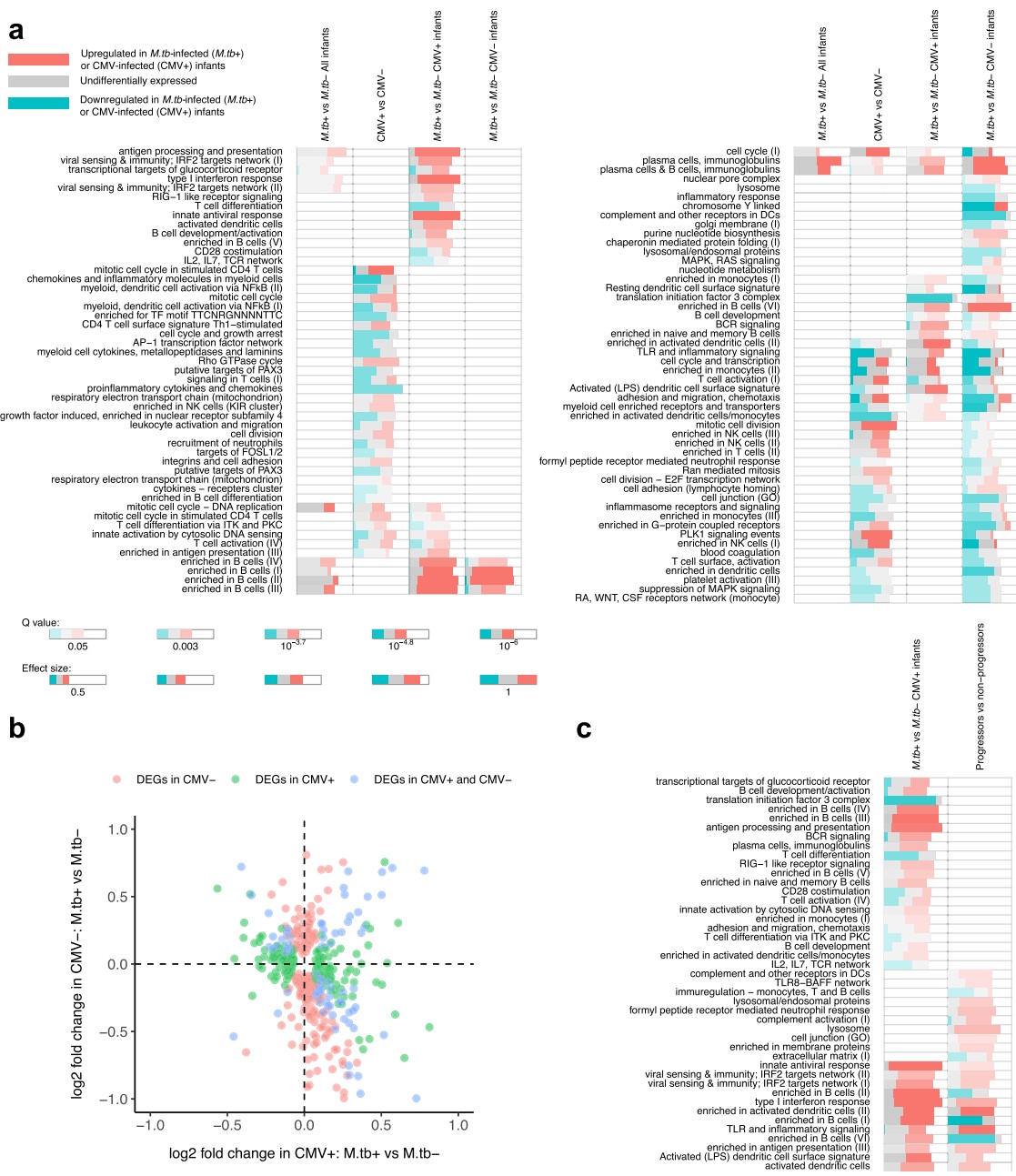

**Fig. 4 | Gene set enrichment results using the CERNO algorithm and gene sets defined by Li et al.[20]. a** The first, third and last columns show enriched gene sets for differential expression between *M.tb*-infected and *M.tb*-uninfected infants among all, CMV-infected and CMV-uninfected infants respectively. The second column shows enriched gene sets for differential expression between CMV-infected and CMV-uninfected infants. Each row represents a gene set. The size of the rug corresponds to the AUROC value in the CERNO test and the colour intensity of the rug corresponds to the enrichment q value. The proportion of red, blue and grey indicates the proportion of upregulated, downregulated and not differentially expressed genes within the gene set, respectively. Only gene sets with AUROC values larger than 0.75 are shown here. Other gene sets are listed in Supplementary Data 7. **b** The horizontal axis shows log₂ fold change of gene expression between *M.tb*-infected and *M.tb*-uninfected infants among CMV-infected infants, while the vertical axis shows that among CMV-uninfected infants. Each point represents a gene. The red and green points indicate differentially expressed genes (DEGs) in CMV-uninfected infants and CMV-infected infants, respectively, while the blue points indicate DEGs in both CMV-infected and CMV-uninfected infants. Only parts of the DEGs were shown in the figure. **c** The first column shows enriched gene sets for differential expression between *M.tb*-infected and *M.tb*-uninfected infants among CMV-infected infants in the current study, while the second shows enriched gene sets for differential expression between TB progressor and non-progressor in adolescents in the South African Adolescent Cohort Study (ACS)[7]. Only gene sets with AUROC value larger than 0.75 are shown here. Other gene sets are listed in Supplementary Data 9. Source data are provided as a Source Data file.

## Ag85A-specific IgG response correlates negatively with BCG-specific IFN-γ response

We evaluated the possible associations between the BCG induced antibody response and T-cell response in our infection cohort and found a significant negative correlation between the ELISpot BCG response and Ag85A-specific IgG response on Day 28 in the placebo group. This was the case both when the analysis included the ELISpot BCG response data from Day −7 samples only and when Day −7 and Day 28 data were combined (Fig. 5a, *p < 0.05*). We then classified the infants into BCG-specific IFN-γ responders (SFC ≥ 60) and non-responders, selecting a cut-off (SFC = 60) which can separate the distribution of the BCG-specific IFN-γ response into a high-density region and a low-

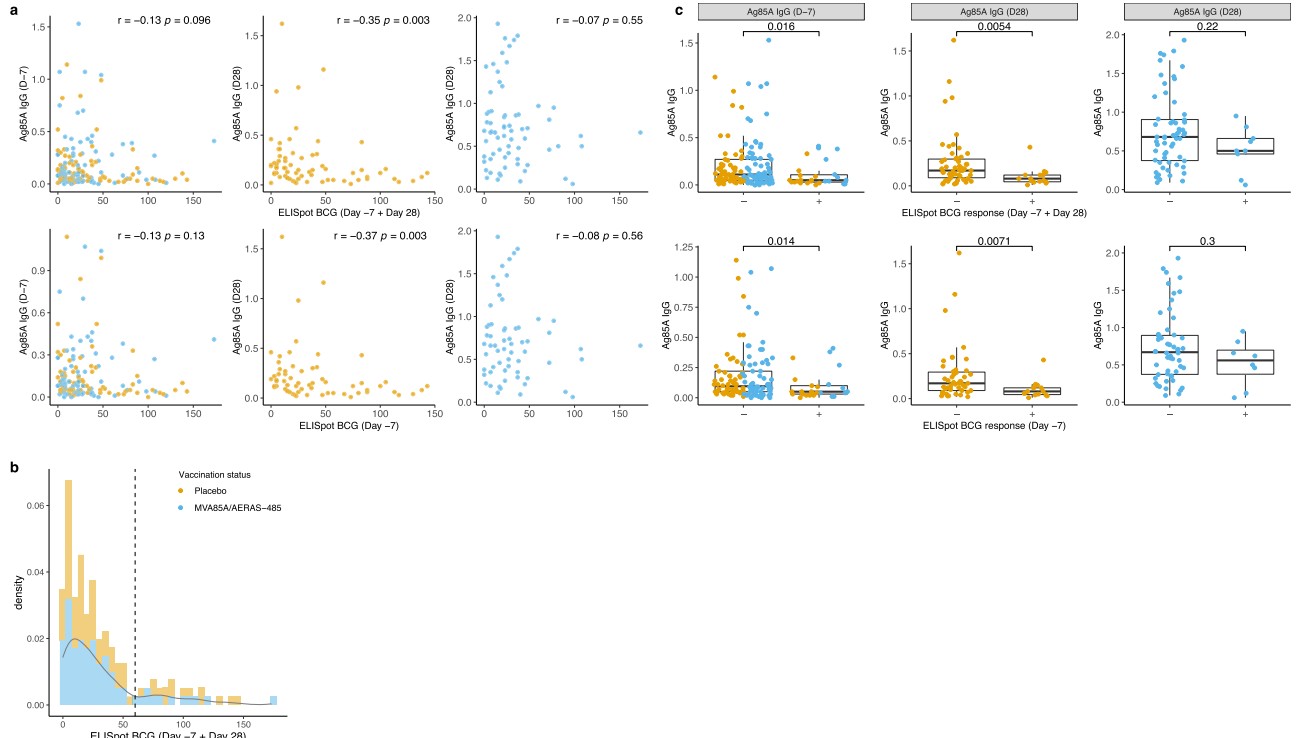

**Fig. 5 | The relationship between Ag85A-specific IgG response and BCG-specific IFN-γ response. a** Spearman Correlation between Ag85A-specific IgG response on Day −7 (Left) or Day28 (middle and right) and BCG-specific IFN-γ response. Day −7 and Day 28 samples are included in the upper panel and only Day −7 samples are included in the bottom panel (Only unadjusted *p*-values are shown). For Ag85A-specific IgG response on Day −7, combined MVA85A-vaccinated and placebo group infants' data were analysed. For Ag85A-specific IgG response on Day 28, MVA85A-vaccinated and placebo group infants' data were analysed separately. Hypothesis testing used to generate *p*-values for the spearman correlation was two-sided. *P*-values were not adjusted. OD values are displayed for antibody response. **b** The distribution of BCG specific IFN-γ response for all samples (both Day −7 and Day 28 samples are included). **c** Ag85A-specific IgG response on Day −7 (Left) or Day28 (middle and right) in BCG T-cell responders (ELISpot BCG response > 60 SFC) and

non-responders (ELISpot BCG response ≤ 60 SFC) (only unadjusted *p*-values are shown). For Ag85A-specific IgG response on Day −7, combined MVA85A-vaccinated and placebo group infants' data were analysed together. For Ag85A-specific IgG response on Day 28, MVA85A-vaccinated and placebo group infants' data were analysed separately. Two-sided Mann-Whitney test without multiple testing correction was used. Bars show medians with interquartile ranges (IQR), the upper whisker extends to the largest value no further than 1.5× IQR from the hinge, the lower whisker extends from the hinge to the smallest value at most 1.5× IQR from the hinge. *N* = 132/26, 120/25, 54/15, 47/15, 59/9 and 54/8 for biologically independent samples from BCG T-cell responders/nonresponders in the left top, left bottom, middle top, middle bottom, right top and right bottom, respectively. OD values are displayed for antibody response. Source data are provided as a Source Data file.

density region (Fig. 5b). The Ag85A specific IgG response on Day −7 was significantly higher in IFN-γ non-responders (Fig. 5c). Whereas on Day 28, the Ag85A-specific IgG response was significantly higher in IFN-γ non-responders only in the placebo group.

## Discussion

In this study we first investigated whether the three immune correlates of risk of TB disease (HLA-DR expression on CD4[+] T-cells, frequencies of BCG-specific IFN-γ in PBMC and plasma Ag85A-specific IgG responses on Day −7) were also associated with susceptibility to *M.tb* infection in BCG-vaccinated South African infants, using the same protocols to ensure comparability of the results obtained from both studies[6]. Findings of the present study showed that correlates of risk of TB disease were not associated with an increased risk of *M.tb* infection in infants who became infected with *M.tb*, but did not progress to active TB during the course of the study. Susceptibility to TB disease and *M.tb* infection may be mediated in part by different immunological mechanisms, which might have implications in designing vaccines targeting prevention of *M.tb* infection and in extrapolating findings from PoI studies to PoD studies.

We evaluated a range of exploratory immune parameters in *M.tb*-infected infants and *M.tb*-uninfected infants, none of which were associated with increased risk of subsequent *M.tb* infection after correcting for multiple comparisons. However, immune parameters

measured by the multiplex assay had much higher statistical significance and higher AUROC values than those measured by other methods. We found upregulation of cytokines, chemokines and complement components such as TNF-α and IP-10, in the plasma when we analysed these parameters separately, indicating that they may mediate, at least in part, the increased risk of *M.tb* infection. As we showed in the results, only 3 out of these 45 immune parameters were associated with CMV infection of the infants, suggesting that the association between most of these immune parameters and risk of *M.tb*-infection were not confounded by CMV infection. The 3 immune parameters associated with both CMV infection and risk of *M.tb*-infection, IP-10, TNF-α and Complement C2, might contribute mechanistically to the increased risk of *M.tb*-infection in CMV-infected infants. Some of these proteins, such as IFN-α2, IP-10 and VEGF, have previously been shown to be correlated with risk of TB disease[22]. Complement Factor I and Complement C5, which were both associated with increased risk of *M.tb* infection, had significantly different profiles at Day −7 compared with Day 28 samples. It is not possible to determine whether the significant results observed for these two factors when the two timepoints were analysed together were confounded by the MVA85A vaccination, late BCG-induced response or just an effect of increased sample size.

For all immune parameters except for Ag85A-specific IgG and *M.tb* H37Rv whole cell lysate-specific IgG responses (for which a more

comprehensive sample set of plasma was available), the majority of the PBMC samples were from Day −7 except when they were not available. All samples used to measure these immune parameters in *M.tb*-uninfected infants were collected on Day −7, whereas one third of the blood samples from infants who went on to be infected with *M.tb* were collected on Day 28 after receiving MVA85A or placebo. As shown in the results, the relationship between these parameters and the risk of *M.tb* infection for most parameters were not confounded by the sample collection timepoint, or, therefore, MVA85A vaccination. For Ag85A-specific IgG and *M.tb* H37Rv whole cell lysate-specific IgG responses, we used both Day −7 and Day 28 samples for each infant when they were available. Day −7 samples were not confounded by sample collection timepoint or MVA85A vaccination. As for the relationships between the responses on Day 28 and risk of *M.tb* infection, we showed in the results that they were also not confounded MVA85A vaccination or sample collection time point because there was no significant difference in either group when samples from the placebo group and the MVA85A group were analysed separately. For the transcriptomic analysis to find differentially expressed genes between *M.tb*-infected and *M.tb*-uninfected infants, Day 28 samples were used in only 2 out of the 21 *M.tb*-infected infants, so we did not assess the confounding of the sample collection time or MVA85A vaccination.

CMV is a common persistent herpesvirus infection that modulates immune functions. CMV infection is highly prevalent in low-and middle-income countries, is known to cause immune activation, and is associated with risk of TB disease[8,11,23,24]. The samples used in this study were from 3-6 month old infants, an age considered to be the peak age of primary CMV infection[25]. We evaluated CMV infection status by measuring CMV-specific IFN-γ responses using a pool of CMV peptides and also by measuring CMV-specific IgG and IgM responses. The goal of measuring CMV-specific IgG responses at two time points and only considering those infants that maintained at least 90% of their initial response and were CMV-specific IgG positive at the later timepoint as CMV-infected was to differentiate exposure to CMV from passive maternal CMV antibody transmission. Measuring CMV-specific IgG was more sensitive than measuring CMV cellular responses to peptides, which was probably limited by the number of peptides used and HLA-restriction.

We observed a tendency towards an increased prevalence of subsequent *M.tb* infection in CMV-infected infants compared to those with negative CMV reactivity. CMV-infected and CMV-uninfected infants have distinct *M.tb* infection signatures; hence, CMV infection status should be taken into consideration when characterising *M.tb* infection gene signatures, in line with our previous observation from the disease cohort[11]. CMV infection induces inflammatory responses that may modulate the immune system, such as the upregulation of TNF-α, IP-10 and complement C2 in the plasma and HLA-DR expression on CD4⁺ T-cells and CD8⁺ T-cells. CMV-induced immune activation might have perturbed the immune system and led to an impairment of BCG-induced immunity, which could explain the trend towards increased numbers of *M.tb* infected infants in the CMV-infected infants. In addition, immune activation may influence early immune defence mechanisms in the airways, which might contribute to higher susceptibility to *M.tb* infection.

As samples from infants who went on to develop TB disease were used in our previous correlates of risk of disease analysis, in this study we have evaluated immune correlates of risk of *M.tb* infection for a subset of infants who did not go on to develop TB disease. Thus these infants could be considered to be protected from TB disease, in parallel with being susceptible to *M.tb* infection. This is particularly true as the majority of infants infected with *M.tb* at Day 336 of the study (13/15) reverted to negative QFT reactivity during the study follow-up period. The immune parameters we have identified here may therefore associate with successful clearance of *M.tb* infection. Further work on

this is needed to gain further insights into protective immunity against *M.tb* infection. Unfortunately, there was not an adequate number of sustained QFT converters to perform any direct comparison with QFT reverters. Study subjects were not followed up beyond end of study date to identify TB disease progressors. It is still unclear whether TB disease incidence is lower in QFT reverters than in those with sustained QFT reactivity. In South African adolescents, TB incidence among QFT reverters was no different from whose with sustained QFT reactivity, and the incidence in both these groups was higher than the TB incidence among sustained QFT negative adolescents[18].

Total RNA-seq analysis revealed a number of genes that were differentially expressed between *M.tb*-infected and *M.tb*-uninfected infants. Unexpectedly, immunoglobulin-related genes were among those upregulated in *M.tb*-infected infants. The upregulation of immunoglobulin genes in *M.tb*-infected infants was found in both CMV-infected and CMV-uninfected infants when we analysed the data separately, suggesting that the upregulation of immunoglobulin genes was not simply a response to the elevated inflammation driven by CMV infection. Levels of Day −7 Ag85A-specific plasma IgG were also higher in infants who became infected with *M.tb* during the follow-up period. The Day −7 Ag85A-specific IgG were unlikely to be maternal, since even in the placebo group, Ag85A-specific IgG response were increased in most infants (72% of 72 infants) on Day 28 compared to that in Day −7 (Supplementary Fig. 5). We have previously shown that although only Ag85A-specific IgG on Day 28 were associated with a significantly reduced risk of TB disease, Ag85A-specific IgG on Day −7 also showed a trend towards an association with reduced risk of TB disease that did not reach statistical significance[6]. Whereas our present study illustrates an association between Ag85A-specific IgG on Day −7 and increased risk of *M.tb* infection, but no association for Ag85A-specific IgG on Day 28. The role of antibodies in protection might be limited to development of TB disease rather than acquisition of *M.tb* infection, further investigations are needed to interrogate the role of antibodies in more detail. Although these immunoglobulin-related transcripts were upregulated in infants who became infected with *M.tb*, none of the infants who converted to positive QFT at Day 336 progressed to active TB disease in the study and the majority of infants reverted to a negative QFT, suggesting these infants cleared their *M.tb* infection. The negative correlation between Ag85A-specific IgG response and BCG-specific T-cell response in this cohort, may suggest a potential skewing of BCG response towards antibody response and away from T-cell response.

The overlap between genes that were differentially expressed between the *M.tb*-infected vs. *M.tb*-uninfected groups and the CMV-infected vs. CMV-uninfected groups was dominated by immunoglobulin-related genes. Within each of CMV-infected and CMV-uninfected infants, these signatures were still higher in *M.tb*-infected infants compared to *M.tb*-uninfected groups, indicating that this effect is not entirely attributable to CMV infection and the antibodies translated from these immunoglobulin genes might not all be CMV-specific.

Transcripts encoding components of the complement system and pathways related to complement components were upregulated in infants who were subsequently infected with *M.tb* in this study. This was coupled with increased plasma levels of complement components such as complement C4b and C2. The complement system is an important part of the innate immune system and functions as a proteolytic cascade. The role of the complement system in TB has been investigated previously, and the complement cascade was found to be elevated prior to TB diagnosis in the ACS cohort previously described[26]. At a molecular level, pathogenic mycobacteria invasion mechanisms require the association of the complement cleavage product C2a with mycobacteria resulting in the formation of a C3 convertase, which leads to C3b opsonization of the mycobacteria and recognition by macrophages[27].

IFN-related transcripts, together with plasma levels of IFN-$\alpha$2 and IP-10, were upregulated in infants who became infected with *M.tb*. This is consistent with previous observations of the association between increased IFN response and risk of TB disease and the presence of increased IFN transcripts in *M.tb*-infected adolescents who progressed to active TB[7,11]. One of the upregulated IFN-related genes is the negative regulator of IFN response (*NRIR*), which is one of the negative regulators that can be exploited by pathogens to escape type-I IFN-mediated antimicrobial activity, a feature that can be utilised to support *M.tb* survival[28]. Modular analysis revealed upregulation of type-I IFN and innate antiviral pathways in *M.tb*-infected infants. In addition, plasma levels of many inflammatory cytokines, such as TNF-$\alpha$, IFN-$\gamma$, IL-12 and GM-CSF, were also upregulated in the *M.tb*-infected infants. A recent nested case-control study using serum samples from TB patients in Uganda prior to a diagnosis of TB showed that the IP-10 level was positively correlated with increased CMV IgG level and the risk of TB disease[12]. In a murine model of chronic lymphocytic choriomeningitis virus (LCMV) and *M.tb* coinfection, chronic infection-induced TNF-$\alpha$ was found to inhibit the induction of *M.tb*-specific T-cell immunity, contributing to the increased burden of *M.tb* in the LCMV and *M.tb* coinfected mice[29].

In our transcriptomic datasets, we found gene sets related to the inflammatory response, such as type I interferon response, viral sensing & immunity, were upregulated in *M.tb*-infected compared to *M.tb*-uninfected infants when we analysed all infants combined or CMV-infected infants alone, but not CMV-uninfected infants. This suggests the upregulation of inflammatory signatures we found in *M.tb*-infected compared to *M.tb*-uninfected infants for all infants combined was mainly driven by CMV infection.

Sample size was limited in this study considering this analysis necessarily focused on the subset of infants that converted in their QFT reactivity, did not develop TB disease nor received Isoniazid Preventive Therapy (IPT). Identification and validation of correlates of risk of *M.tb* infection and confirming whether there are common correlates of risk for both TB disease and *M.tb* infection is essential for the current strategy of using PoI as a surrogate for PoD to accelerate TB vaccine R&D. PoI trials require smaller sample sizes and a shorter duration of follow-up compared to PoD vaccine trials, and are therefore considered useful to provide a biological signal of efficacy prior to committing to PoD trials. The success of this strategy relies on there being common correlates of risk of TB disease and *M.tb* infection. Therefore, despite the limitations of sample size, these samples provided a unique opportunity to identify correlates of risk of *M.tb*-infection and shed lights on using PoI trials as a surrogate for PoD trials, especially in infants.

In this study, we excluded infants who received IPT, which might include infants who were QFT-positive and did not progress to TB disease but received IPT because they were also Tuberculin Skin Test (TST)-positive. For the infants included in our study, we did not know their TST status; it is therefore possible that these infants were not representative of all QFT-positive infants.

Our study identifies that inflammation and immune activation are associated with risk of *M.tb* infection in a subset of BCG-vaccinated South African infants who converted to positive QFT but did not progress to active TB disease. We have also shown that our previously identified correlates of risk of TB disease are not associated with susceptibility to *M.tb* infection in this cohort. Further work on these and other clinical trial cohorts is needed to understand the underlying immunological and molecular pathways in order to inform vaccine design and development.

## Methods
### Study design and volunteers
Samples for this study were collected from 3-6 month old, HIV-negative South African infants who received BCG within 7 days of birth and participated in the MVA85A efficacy trial (ClinicalTrials.gov number NCT00953927[5]). Work described in this manuscript is covered by informed consent, which was obtained from the mothers of all infants and by Ethical approval obtained from the University of Cape Town Faculty of Health Sciences Human Research Ethics Committee, Oxford University Tropical Research Ethics Committee and the Medicines Control Council of South Africa[5]. Information concerning subject recruitment, payment or compensation procedures, etc., was submitted to the Institutional Review Board (IRB) and Independent Ethics Committee (IEC) by the investigator. Mothers received reimbursement for time and inconvenience. All transport was provided to and from the clinic.

Infants without known TB exposure and who tested negative with QFT test were randomized to receive a single intradermal dose of MVA85A or placebo. *M.tb* infection was defined as QFT test conversion from negative to positive at the manufacturer's threshold of 0.35 IU/mL IFN-$\gamma$. *M.tb*-uninfected infants were those who remained QFT negative and were included in the study at a ratio of 3 *M.tb*-uninfected to 1 *M.tb*-infected. *M.tb*-uninfected controls were matched based on sex, ethnic group, Centre for Disease Control (CDC) weight-for-age percentile and time on study. Infants who lived in a household with a smear-positive TB case or received IPT were excluded from this study, and none of the infants included in this study had active TB disease during the study follow-up period.

Blood samples, collected at baseline, from 43 infants that subsequently met the criteria for *M.tb* infection, according to the study protocol, were used in this study;[5] 17/43 infants were from the group that received MVA85A (Fig. 6). Samples for this study were mainly from Day −7 of the study (7 days before BCG-vaccinated infants were randomized to receive vaccination with either MVA85A or placebo). Fourteen infants did not have Day −7 PBMC samples and in these infants Day 28 post-vaccination samples were used; six of these infants were in the MVA85A group and eight were in the placebo group. Seventeen infants did not have Day −7 plasma samples, hence Day 28 post-vaccination samples from these infants were used instead. Eight of these infants were in the MVA85A group and nine were in the placebo group. Blood samples from all 127 matched *M.tb*-uninfected controls were from Day −7 of the study. 69/127 were from infants that received MVA85A.

### Sample processing
Immune assays were performed on frozen PBMC that were thawed and rested for 2 h in a 5% $CO_2$ incubator at 37 °C as previously described[6].

Considering the large number of samples to be run, they were processed in 6 batches, we ensured that samples from each case and their corresponding matched controls were included in the same run. The assays performed included, in priority order: flow cytometry surface staining for characterisation of T-cells, ELISpot assay, flow cytometry surface staining for characterisation of B-cells, MGIA, gene expression and flow cytometry surface staining for characterisation of MAIT cells. The laboratory researchers performing the assays were blinded to the group allocation. Multiplex Assay and ELISA (IgG against Ag85A and whole cell lysate) were performed on all available plasma samples. Supplementary Data 10 summarises the assays performed and the percentage of samples that had all assays and conditions performed in each batch. Immune assays were performed in line with those used in the previous correlates of risk of TB disease study[6,30].

### Gene expression
**Cell stimulation.** Cells were stimulated in RMPI with 10% Foetal calf serum (FCS) at $5 \times 10^6$/ml with $1 \times 10^6$ Colony Forming Units (CFU)/ml BCG (SII) or left unstimulated in a total volume of 200 µl/well in 96-well plates. Cells were incubated for 12 h (37°C, 5% $CO_2$). Plates were centrifuged and 200 µl supernatant was removed without disrupting the pellet and transferred into clean U-bottom plates. The plates were

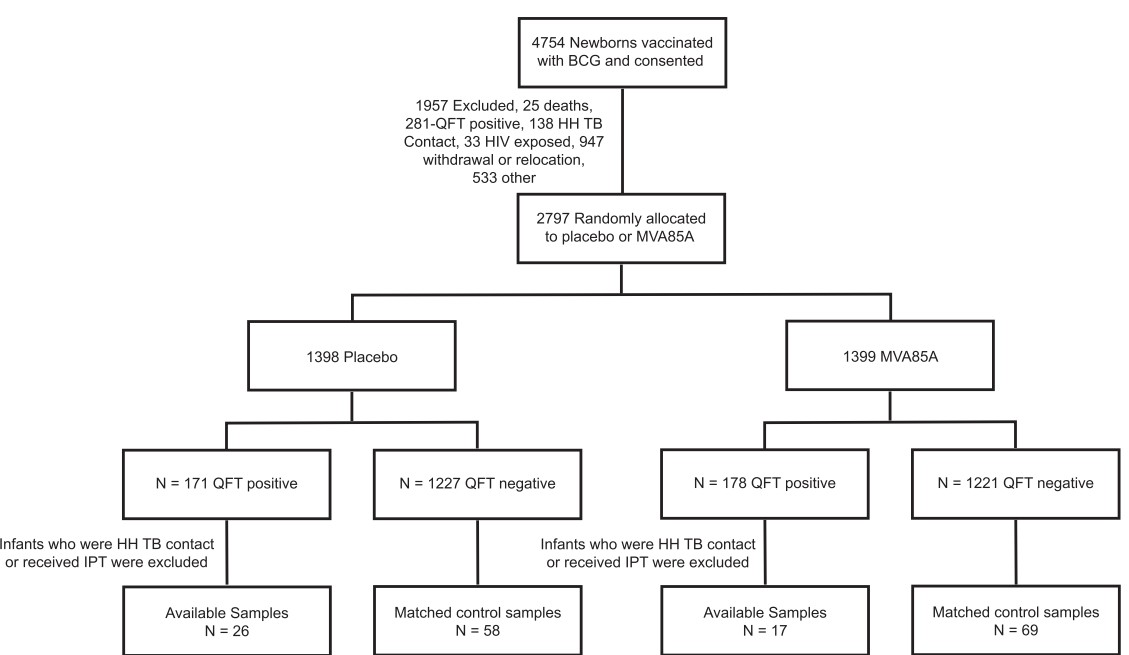

**Fig. 6 | Flow chart illustrates the number of samples from QFT positive (IFN-γ > 0.35 IU/ml) and QFT negative infants.** All *M.tb*-infected infants had 3 matched *M.tb*-uninfected controls based on sex, ethnic group, CDC weight-for-age percentile (±10 points), and time on study (±9 months). HH: Household.

centrifuged again, the supernatant flicked-out and the cells resuspended in 200 µl RLT buffer with β-mercaptoethanol (10 µl/ml). The samples were mixed with the buffer to lyse the cells and the plates were sealed and stored at −20 °C.

At the time of RNA extraction, plates were thawed at 37 °C for 10 min. RNA extraction was performed using the Qiagen RNeasy kit following the manufacturer's instructions (including optional DNase digestion) and RNA was eluted twice, 30 µl per elution. RNA was quantified by nanodrop and samples were stored at −80 °C.

**RNA-Seq.** Total RNA quantity and integrity were assessed using Quant-IT RiboGreen RNA Assay Kit (Invitrogen, Carlsbad, CA, USA) and Agilent Tapestation4200. Samples were normalised to 70 ng input. Purification of mRNA, generation of double stranded cDNA and library construction were performed using NEBNext Poly(A) mRNA Magnetic Isolation Module (E7490) and NEBNext Ultra II Directional RNA Library Prep Kit for Illumina (E7760L) with adapters and barcode tags (dual indexing)[31]. The concentrations used to generate the multiplex pool were determined by Picogreen. The final size distribution of the pool was determined using a Tapestation system (Agilent), and quantification was determined by Qubit (Thermofisher) before sequencing on an Illumina HiSeq 4000 as 75 bp paired end.

**Gene expression data analysis.** Quality control checks on raw sequence data were performed using fastqc (v0.11.8) (http://www.bioinformatics.babraham.ac.uk/projects/fastqc/). Reads were aligned to the genome using HISAT2 (v2.1.0)[32]. Sequences were quantified as gene counts using the featureCounts (subread v2.0.2) tool[33]. A linear model was fitted using DESeq2 (v1.22.2) to determine differentially expressed genes (DEGs) adjusted for matching group and stimulus[34]. Genes with an adjusted *p*-value of less than 0.2 and a fold change of more than 5% were considered DEG.

Pathway analysis was performed using the Coincident Extreme Ranks in Numerical Observations (CERNO) algorithm from tmod package (v0.40) in R[35]. Modules used for enrichment analysis were adapted from Li et al.[20]. Modules with an adjusted *p*-value of less than 0.001 were considered enriched in the analysis.

GO enrichment analysis[36] was performed using the enrichGO function in clusterProfiler package (v3.10.1) in R (v 4.1.2)[37]. DEGs were grouped into upregulated genes and downregulated genes, and then enriched separately. The get_child_nodes function in GOfuncR package (v1.2.0) in R was used to obtain all child GO terms of immune system process (GO:0002376) and cytokine production (GO:0001816). Then, all enriched GO terms that are not child GO terms of immune system process (GO:0002376) and cytokine production (GO:0001816) were removed. If the number of remaining GO terms of a group was larger than 50, we only kept the most significant terms among terms with similar semantics (cut-off = 0.7) using the simplify function in the clusterProfiler package, which is measured by a similarity measure introduced by Wang et al.[38] and calculated using GOSemSim (v2.16.1) package in R[39]. Supplementary Fig. 4 was drawn using simplifyEnrichment package (v.1.0.0) in R with cutoff set as 0.75[40]. Downregulated gene sets in infants who were subsequently *M.tb*-infected, compared to *M.tb*-uninfected (All infants) and downregulated gene sets in infants who were subsequently *M.tb*-infected, compared to *M.tb*-uninfected (CMV-infected infants) are not shown because the number of enriched gene sets in these two situation is less than 2.

**Enzyme-linked immunoSpot (ELISpot) assay**
IFN-γ ELISpot was performed as previously described[6]. $3 \times 10^5$ PBMC/well were stimulated in duplicate with each of $2 \times 10^5$ CFU/ml BCG SII, 2 µg/ml Cytomegalovirus (CMV) peptides and 2µg/ml Epstein-Barr Virus (EBV) peptides (both from AnaSpec; CMV = 5 peptides (4 from pp65, 1 from IE1) and EBV = 15 peptides), 20 µg/ml purified protein derivative (PPD) (AJ Vaccines), 10 µg/ml phytohaemagglutinin (PHA) (SIGMA) or left unstimulated as negative control. Plates were read in an ELISpot reader (AID ELISpot (v7.0 iSpot). Antigen specific responses are presented as Spot Forming Cells (SFC)/$1 \times 10^6$ PBMC, calculated by subtracting the mean of the unstimulated responses from the mean of the antigen-specific responses and correcting for the number of PBMC per well. A response was considered positive if the mean number of spots in the antigen well was at least twice the mean of the unstimulated wells and at least 5 spots or greater.

### Enzyme-linked immunosorbent (ELISA) assay

Ag85A-specific IgG was measured as previously described[6]. Plates and reagents were brought to room temperature and plates were incubated sealed. ELISA plates were coated with 50 ml of 2 $\mu$g/ml rAg85A (BEI resources) diluted in Sodium Carbonate buffer and were incubated for overnight at 4 °C. Plates were washed three times with Phosphate-Buffered Saline (PBS), 5% Tween-20 (v/v) and were then blocked for 1 h at room temperature with 100 ml blocking buffer (PBS, 5% Milk Blocking (w/v)). Serum samples were diluted 1/100 with PBS, 5% Milk Blocking (w/v) and 50 μl of each diluted sample was plated in triplicate. Controls and plate blanks consisting of assay diluent alone with no serum were added in triplicate. Plates were incubated for 2 h at room temperature, washed, 50 μl of goat anti-human (KPL) detection antibody (1:500 dilution) was added to each well and plates were incubated for 1 h at room temperature. 50 μl TMB (BD) was added to each well and plates were incubated for 15 min in the dark before the colour developing reaction was stopped by adding 50 μl of 2 M Sulfuric Acid (Sigma) to each well. Absorbance was measured using a microplate reader at 450 nm using GEN5 software (v2.07). Results are presented as mean OD values with the mean OD values of the blank wells (without plasma) subtracted.

### CMV reactivity

Evaluation of CMV infection was performed by detection of a cellular immune response to CMV peptides by ELISpot and by detection of CMV-specific IgG and IgM responses at Day −7 and Day 28, using commercial kits (Human Anti-Cytomegalovirus IgG ELISA Kit, Abcam) and following the manufacturer's instructions. The absorbance of samples was transformed to standard units according to the manufacturer's recommendation. Plates were read in an ELISA reader using GEN5 software (v2.07). Samples with standard units greater than 11 were considered positive, whereas those with standard units of less than 9 were considered negative. The assay was repeated for samples with standard units of between 9 and 11, which were considered inconclusive. If the results of the second test were less than 11, then the samples were considered negative. Otherwise, they were considered positive. CMV-specific IgG responses were only evaluated in the infants from whom both Day −7 and Day 28 samples were available. For CMV-specific IgM responses, we initially tested Day 28 samples, and tested Day −7 samples where Day 28 samples were not available.

### Flow cytometry surface staining

Surface staining of PBMC was performed as previously described[6,30]. Cells were stained with antibodies to characterize T, B and MAIT-cells (Supplementary Data 11). To ensure assay reproducibility; antibodies from the same lot were used for staining all samples. Staining was performed in V-bottom 96-well plates, cells were washed in PBS, viability marker diluted in PBS was added to cells, which were then incubated for 10 min at 4 °C. Surface antibody mix was added and cells were incubated for 30 min at 4 °C. Cells were then washed in FACS buffer (PBS, 1% BSA (Sigma) and 0.1% Sodium Azide (Sigma)). Cells were resuspended in 100 μl of 1% Parafolmaldehyde (Aesar) / FACS buffer and were acquired on an LSRII (BD Biosciences), using FACS DIVA (v6.2) and data analysed using Flowjo v8.8 (BD Biosciences). Dead and doublet cells were excluded from the analysis. Fluorescence-Minus one (FMO) gating controls were used. Gating strategies for the 3 panels are illustrated in Supplementary Fig. 6.

### Mycobacterial growth inhibition assay (MGIA)

PBMC ($3 \times 10^6$) were incubated with 500 CFU BCG Pasteur and 120 μl autologous serum in a total volume of 480 μl RPMI/well in a 48-well tissue culture plate. Cultures were incubated at 37 °C for 96 h. Cells were then transferred to 2 ml screw-cap tubes and centrifuged at 15294 g for 10 min followed by removal of the supernatant. 500 μl sterile water was added to the wells to lyse adherent monocytes and release intracellular BCG, and was then added to the corresponding pellet in the 2 ml tubes. The 2 ml tubes were pulse-vortexed and the lysate was transferred to BACTEC MGIT tubes supplemented with BBL MGIT OADC and PANTA (Becton Dickinson, UK). MGIT tubes were placed on the BACTEC 960 machine (Becton Dickinson, UK) and incubated at 37 °C until the detection of positivity by fluorescence. The time to positivity (TTP) read-out was converted to $\log_{10}$ CFU using stock standard curves of TTP against inoculum volume and CFU. Control tubes were set up at day 0 by inoculating supplemented BACTEC MGIT tubes with the same volume of mycobacteria as the samples. Results are presented as normalised mycobacterial growth which is equal to ($\log_{10}$ CFU of sample−$\log_{10}$ CFU of growth control).

### Multiplex

A bead-based multiplex assay to analyse a range of 38 Th1 and Th2 cytokines, chemokines and complement factors was performed on plasma samples. The MILLIPLEX MAP Human Cytokine/Chemokine Magnetic Bead Panel was used following the manufacturer's recommendations. Data was acquired using xPONENT (v4.2.1705.0) and analysed using drLumi (v0.1.2)[41]. Details of the measured cytokines and chemokines are included in Supplementary Data 10a.

### Statistical analysis

A statistical analysis plan was prepared prior to data analysis in which the methods of analyses were specified and the primary and secondary analyses were defined (Supplementary Information). The primary objective of this study was to identify correlates of risk of M.tb infection in BCG-vaccinated, HIV-uninfected infants and in particular, to determine whether the three markers that were associated with risk of M.tb disease were also associated with risk of M.tb infection[6]. Accordingly, the primary objective was to assess the ability of a set of three selected variables, activated CD4+ T-cells, BCG-specific IFN-γ and Ag85A-specific IgG to predict subsequent M.tb infection during study follow-up in a set of 43 M.tb-infected infants with 3 matched M.tb-uninfected controls per M.tb-infected infant. An exploratory objective was to identify a secondary set of variables measuring the immune response at Day −7 or Day 28 that best classified participants in terms of their risk of M.tb infection; these variables are shown in Supplementary Data 2. A series of univariate conditional logistic regression analyses were performed for each assay to assess the association between Day −7 or Day 28 immune response and risk of M.tb infection. In these analyses, the occurrence of M.tb infection was the outcome variable with the matched case-control set number as the strata variable and the Day −7 or Day 28 immune response as the independent variable. Results are summarized as a list of all immune response variables together with their estimated odds ratios, 95% confidence intervals, and two-sided p-value. Benjamini-Hochberg adjusted p-values (False Discovery Rate, FDR) are presented and an FDR of less than 0.20 was considered evidence of a significant association. Furthermore, the area under the receiver operating characteristic (AUROC) curve was calculated for each immune response. Immune responses with an AUROC of between 0.80 and 0.90 were considered to be 'good' discriminators of infants who were at risk of going on to become M.tb infected, while responses with an AUROC of greater than 0.90 were to be considered 'excellent' discriminators. All adjusted p-values shown in this paper are adjusted by Benjamini-Hochberg multiple testing correction. All statistical tests done in this work were two-sided. Statistical analysis was done using STATA (v14) and R (v4.1.2).

### Reporting summary

Further information on research design is available in the Nature Research Reporting Summary linked to this article.

## Data availability

The RNA-Seq datasets generated during in this study have been deposited in the Gene Expression Omnibus database under accession code GSE203395.

The transcriptomic data in ACS used in this study are available in the Gene Expression Omnibus database under accession code GSE79362.

Source data are provided with this paper.

The immunology datasets generated in this study are provided in the Supplementary Data 12. Source data are provided with this paper.

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

## Acknowledgements

This research was funded by TBVAC2020 and the Wellcome Trust. H.McS. is a Wellcome Trust Investigator (grant code WT 206331/Z/17/Z). For the purpose of open access, the author has applied a CC BY public copyright license to any Author Accepted Manuscript version arising from this submission. Bioinformatics support was provided by the VALIDATE network. We acknowledge the help we received from: Ashley Hockham and Robert Hardcastle (Merck Millipore), Jennifer Hill from (Oxford University and Krista E.van_Meijgaarden (University of Leiden, the Netherlands) for helping with the multiplex assay and data analysis and Naomi Bull for helping with sample processing. We also acknowledge the Oxford Vaccine Group biobank.

## Author contributions

Conception: I.S., R.W., S.L., S.A.H., R.T., H.A.F., T.J.S., M.T., M.H., H.McS. Design: I.S., R.W., S.L., S.A.H, R.T., D.C., N.W., H.M., H.A.F., T.J.S., M.T., M.H., H.McS. Analysis: I.S., R.W., S.L., S.A.H., R.T., D.C., N.W., H.McS., A.J. Data Acquisition: I.S., R.W., S.L., S.A.H., R.T., A.J. Drafting and Reviewing Manuscript: All.

## Competing interests

H.M. was previously a shareholder in the Oxford-Emergent Tuberculosis Consortium (OETC), a joint venture established for the development of MVA85A (OETC no longer exists). The remaining authors declare no competing interests.
