## [Peer Review File · Nature Communications]

Inflammation and immune activation are associated with risk of *Mycobacterium tuberculosis* infection in BCG-vaccinated infantsREVIEWER COMMENTS

Reviewer #1 (Remarks to the Author):

In this well written and thoughtful manuscript, investigators utilize samples collected from the previous MVA85A TB vaccine trial in young South African infants to address the question of whether or not previously identified immune correlates of risk for TB disease are also predictive of risk of Mtb infection in this population. The authors put forth that the significance of this question relates to evaluation of future candidate TB vaccines, as studies that focus on prevention of Mtb infection can be performed faster and with smaller sample size than those that focus on prevention of TB disease. Although an extensive number of investigations were performed using samples from infants who did or did not acquire Mtb-infection (defined here by Quantiferon-conversion), the primary analysis focuses on three previously identified immune correlates of TB disease risk: CD4+ T cell activation; BCG-specific IFN-g, and Ag85A-specific IgG. The investigators' attention to CMV status, and the confounding nature of CMV infection on immune activation and its potential role in driving risk of TB, is well appreciated. Although I applaud the use of these samples from the MVA85A trial to try to gain insights into the immunobiology of Mtb-infection in highly vulnerable infants, I have several major concerns that limit enthusiasm about this manuscript and its findings.

First, the sample size is limited and confounded by inconsistencies in blood draw timing and sample availability, receipt of MVA85A vaccine, and heterogeneity in QFT-responses (eg., high incidence of reverters, detailed further below). Of 43 infants who met study definition of Mtb-infection, the timing of blood draws with regards to receipt of MVA85A vaccine versus placebo during trial was inconsistent (14 infants has samples from +D28 while remainder were -D7 for PBMC; this was 17 infants for plasma). Among children who did not meet the study definition for Mtb-infection, all samples were from -D7. Although investigators attempt to provide reassurance that timing of sample collection did not have a significant impact on immunologic parameters (lines 329-345) this was only presented for the cytokine analysis and not the three stated primary immunologic variables. Although it is understandable that there may not be sufficient sample available to complete all planned investigations (as detailed in Supplemental table #1), in some cases only 50% of the study population had sample available further limiting statistical power. Moreover, it is noted that this current analysis is a sub-analysis of a much larger vaccine trial. In the original trial, sample size was selected so that differences in exposure to Mtb and potential confounders could safely be assumed to be equal between groups assigned to vaccine vs placebo. Here, with a much smaller sample size, how can we be reassured that environmental factors that could impact the outcome (Mtb-infection) or investigated assays of immune activation, are equally distributed between the two groups? Indeed, an important finding of this analysis is the critical role of CMV in shaping immunologic outcomes in young infants; however, this adds a further layer of complexity to data interpretation. Although the authors have attempted to tease out the impact of CMV on the study primary outcomes, this further limits the sample sizes being compared, the statistical power, and confidence in the presented results. I have major concerns about the reliability of the statistical conclusions and reproducibility of the findings.

Another major concern is the transient nature of the study outcome. Specifically, 14 of the 43 children who were identified as Mtb-infected based on QFT conversion from negative to positive during the study period, reverted to QFT negative by the end of study (Figure 2). Thus, for a large number of infants included in this analysis, they only transiently met the definition of Mtb-infection. The clinical significance of QFT reversion remains unknown as the authors acknowledge; however, there is either something immunologically distinct among the children who reverted or their initial QFT conversion was borderline initially and perhaps reflected a false-positive result. Either way, this is extremely problematic for an immune-based analysis where the sample size is too small for the reverters to be considered separately, and further confounds the results.

Regarding the significance of the question this study was designed to address—do immunologic risk factors for Mtb infection align with those identified for TB disease—to this reviewer, there is limited rationale to support this line of investigation. The findings of this study, although they may be of interest to those in the field of TB vaccine development, seem rather predictable.

Several additional questions:

1) The Gene expression data analysis applies liberal definitions for DEGs, both with regards to the thresholds for a significant adjusted p-value (less than 0.2) and fold change (defining a fold change as 5% difference or more). What is the justification for these definitions? Related to this, in Figure 5A, the legend states that q-values are shown, however in the actual figure it states p-value. This should be clarified.

2) Were HIV-exposed-uninfected infants included in this analysis? In the parent vaccine trial, infants had to be HIV ELISA negative, but this does not preclude HIV perinatal exposure as maternal HIV antibodies could be undetectable by age 4-6 months. Given concerns about abnormal immune function among HEU infants in the first year of life, as well as increased risk of Mtb-exposure among HEU infants, this should be considered. If maternal HIV testing was performed during the parent trial, these data could be used to address this consideration.

3) Is there a possibility that the plasma Ag85A-specific IgG responses detected on -D7 were maternal in origin? How was this excluded as a possibility? If this was not excluded, how do the authors interpret the findings?

4) The formatting for figure 3A is confusing and appears to contain unadjusted/uncorrected statistics. I am having difficulty understanding the two sets of p-values presented (y-axis versus overlaid).

5) The data presented in Figure 6 is difficult to understand given receipt of MVA vaccine by many study participants. What is the significance of these findings?

6) Please review references, I noted several citations that appeared incorrectly placed.

Reviewer #2 (Remarks to the Author):

The vast majority of individuals exposed to Mtb do not develop active tuberculosis disease, even if there is immunological evidence (by IGRA) of probable infection. Indeed, evidence from studies of healthcare workers in the preantibiotic era suggests that tuberculin skin test positivity, if not associated with progression to active TB, was actually protective against subsequent TB disease.

Furthermore, since they are easier and cheaper to run, there has been an interest in “prevention of infection” trials for candidate TB vaccines compared with “prevention of disease” trials. The assumption of these efficacy trials, is of course, that in an efficacious POI vaccine, those prevented from getting “infected” (assuming <100% efficacy) overlap with those that would develop TB disease. Investigating that assumption lies at the heart of this potentially important study from McShane and colleagues. The bottom line they present is that this may be an over-simplification.

The authors mine the rich clinical samples from the failed MVA85A candidate TB vaccine trial. Infants were all BCG vaccinated and then randomized to MVA85A or placebo. In the original study, ~70 infants developed active TB disease, split fairly evenly between both arms. In a post-hoc analysis, several features were identified to be potential correlates of protection against ATB: including HLA-DR upregulation on CD4+ T cells (risk of ATB), BCG-responsive ELISPOT and Ag85A-specific IgG responses _28 days post-MVA-A85A/placebo_.

A major strength of the study is that the infants defined as “infected” were rigorously evaluated as to not developing ATB over the course of the study in a naturalistic fashion. The authors examine the cohort of infants that were “infected” with Mtb (as defined by a single positive IGRA), but importantly, did not develop ATB during the course of the study AND did not receive isoniazid preventative therapy (IPT) [which would be expected to decrease risk of subsequent ATB in IGRA+ infants]. Thus these infants could be confidently assigned as not developing ATB in a naturalistic fashion during the

course of the study. However, household contacts of ATB individuals were excluded from the analysis. It is not clear if this is because all HH contact infants received IPT [as guidelines suggest they should] or for some other reason. This is a shame, since if there were infants that were HH contacts and did not receive IPT and did not develop ATB, they would offer the strongest evidence of infection without ATB.

A surprising observation from the study was that Ag85A-specific IgG responses (pre MVA85A/placebo) was associated with increased risk of Mtb infection but not disease. Less surprisingly, general immune activation was associated with risk of Mtb infection. Ag85A-IgG responses were negatively correlated with BCG-specific ELISPOT, suggesting that some infants respond to BCG divergently in terms of humoral vs. cell-mediated responses. Overall, this is a very important study. However, there are a few nuances in terms of interpretation of the data that should be addressed by the authors.

Comments

1. A major point of discussion in the results is that in a former study (Fletcher et al 2016), Ag85A-specific IgG was anti-correlated with ATB and here it is correlated with Mtb infection but not ATB. But in that study, the Ab responses were defined post-vaccination, whereas here they are defined as pre-vaccination (i.e. in response to BCG). I'm not sure, but I don't think the D-7 Ag85A responses were protective against subsequent ATB – the authors should clarify. As such, the two studies were in reality studying different things: on the one hand, antibody responses to a subunit vaccine, and in the other, antibody responses to an attenuated live vaccine. As the other analyses show (i.e. B cell markers in the CMV-Mtb+ group), the risk of Mtb infection (but not disease) may be more reflective of the “skewed” response to BCG. This specific type of response (which is extremely unlikely to be limited to Ag85A) may or may not have been associated with protective against subsequent ATB, and requires clarification. As such, the wording e.g. lines 471-4 may need to be more nuanced than they currently are.
2. Somewhat related, in the placebo arm, were D28 (i.e. after placebo) Ag85A-IgG titres positively correlated with subsequent “Mtb infection”?
3. A limitation of the study is that “Mtb infection” is defined by IGRA positivity at a single time-point in the vast majority of the infants. It's unclear to this reviewer what this means and whether it actually represents Mtb infection or instead Mtb exposure that did not establish infection (the field doesn't really know). For example, the major “POI” trial to date is the BCG revaccination study (Nemes et al, NEJM, not cited). In that trial, BCG revaccination was not protective against a single positive IGRA, but it was protective against “sustained” IGRA positivity. It can't be helped that in this study, only two infants had sustained IGRA positivity, so the authors don't have a choice other than to go with a single positive, but this limitation really needs to be explored more in the Discussion and discussed specifically in the context of the Nemes study. It may also warrant rewording of the title and abstract, replacing “Mtb infection” with “positive IGRA”.
4. I am not familiar with South African TB guidelines, but generally, infants and children <5 years should be offered IPT as routine if they have a positive IGRA (or TST). If this is the case, the cases (“Mtb infection”) in this study may represent a biased/non-random sample of children within this prospective cohort. The authors should clarify the guidelines and discuss as appropriate.
5. Line 538. The authors state “we have previously shown that Ag85A-IgG was associated with reduced risk of disease” but they erroneously cite the Lu 2016 Cell paper (that did not show this) instead of the Fletcher 2016 study. This should be corrected.

Reviewer #1 (Remarks to the Author):

In this well written and thoughtful manuscript, investigators utilize samples collected from the previous MVA85A TB vaccine trial in young South African infants to address the question of whether or not previously identified immune correlates of risk for TB disease are also predictive or risk of Mtb infection in this population. The authors put forth that the significance of this question relates to evaluation of future candidate TB vaccines, as studies that focus on prevention of Mtb infection can be performed faster and with smaller sample size than those that focus on prevention of TB disease. Although an extensive number of investigations were performed using samples from infants who did or did not acquire Mtb-infection (defined here by Quantiferon-conversion), the primary analysis focuses on three previously identified immune correlates of TB disease risk: CD4+ T cell activation; BCG-specific IFN- γ , and Ag85A-specific IgG. The investigators' attention to CMV status, and the confounding nature of CMV infection on immune activation and its potential role in driving risk of TB, is well appreciated. Although I applaud the use of these samples from the MVA85A trial to try to gain insights into the immunobiology of Mtb-infection in highly vulnerable infants, I have several major concerns that limit enthusiasm about this manuscript and its findings.

First, the sample size is limited and confounded by inconsistencies in blood draw timing and sample availability, receipt of MVA85A vaccine, and heterogeneity in QFT-responses (eg., high incidence of reverters, detailed further below). Of 43 infants who met study definition of Mtb-infection, the timing of blood draws with regards to receipt of MVA85A vaccine versus placebo during trial was inconsistent (14 infants has samples from +D28 while remainder were -D7 for PBMC; this was 17 infants for plasma). Among children who did not meet the study definition for Mtb-infection, all samples were from -D7. Although investigators attempt to provide reassurance that timing of sample collection did not have a significant impact on immunologic parameters (lines 329-345) this was only presented for the cytokine analysis and not the three stated primary immunologic variables. Although it is understandable that there may not be sufficient sample available to complete all planned investigations (as detailed in Supplemental table #1), in some cases only 50% of the study population had sample available further limiting statistical power.

We agree that the sample size was limited considering this analysis necessarily focused on the subset of infants that converted in their QFT reactivity, did not develop TB disease nor received IPT. Sample size for the MVA85A trial was determined based on the trial's primary readout; work presented in the manuscript is secondary exploratory analysis done on existing samples. We have added some text to the manuscript to acknowledge the limitation of sample size in Line 627-637.

We have performed statistical analysis on effect of time-point (D-7 vs D28) for all studied variables, but in the figure, we only showed those which were significantly different. As described, full results of this analysis are presented in Supplementary Table 6. For ELISpot BCG and activated CD4⁺ T cells included in the primary analysis, their ORs when only D-7 samples are included are in the same direction as that when all

samples (D-7 and D28) are included, and the parameters remain non-significant as correlates of risk of M.tb infection, which suggests that these results are not confounded by including samples from both time points. For Ag85A-specific IgG in the primary analysis, this analysis was performed using samples from the two time points separately, so was not confounded by time point. For cytokines/chemokines/complement, as was mentioned in the Results, we found that only two of the studied variables (Complement Factor I, Complement C5, also seen in Supplementary Figure 1, Supplementary Table 7) are confounded by including samples from both time points. We have added a more detailed discussion about this in Line 342-356 and 504-513.

Regarding the potential for confounding caused by MVA85A vaccination, this is addressed by our discussion of blood drawing timing, since our analyses are mainly based on D-7 sample and as discussed in the last paragraph, only two of the studied variables are confounded by including samples from both time points. We have added a more detailed discussion in Line 515-526.

The influence of QFT heterogeneity is discussed below in response to the point beginning “Another major concern is the transient nature of the study outcome.”.

Moreover, it is noted that this current analysis is a sub-analysis of a much larger vaccine trial. In the original trial, sample size was selected so that differences in exposure to Mtb and potential confounders could safely be assumed to be equal between groups assigned to vaccine vs placebo. Here, with a much smaller sample size, how can we be reassured that environmental factors that could impact the outcome (Mtb-infection) or investigated assays of immune activation, are equally distributed between the two groups? Indeed, an important finding of this analysis is the critical role of CMV in shaping immunologic outcomes in young infants; however, this adds a further layer of complexity to data interpretation.

In this study we mirrored the correlates of TB disease study (Fletcher et al., 2016), and matched the cases (QFT converters) with controls based on gender, ethnic group, CDC weight-for-age percentile and time on study, at a ratio of 3 controls:1 case. In such studies it is impossible to fully control for exposure to all potential infections, but this can be minimised by ensuring case:control matching, use of 3 controls for each case and that all infants lived in the same geographical location with comparable levels of exposure to various infections.

Although the authors have attempted to tease out the impact of CMV on the study primary outcomes, this further limits the sample sizes being compared, the statistical power, and confidence in the presented results. I have major concerns about the reliability of the statistical conclusions and reproducibility of the findings.

The MVA85A trial was the first infant TB vaccine efficacy trial since BCG was tested in infants more than 50 years ago. The trial was also the first to provide samples that would allow studies such as the one presented here. Other more recent TB vaccine efficacy trials, e.g. Nemes et al, NEJM 2018; Tait et al, NEJM 2019; were conducted in adults/adolescents where a much higher rate of CMV infection is to be expected. Therefore, despite the limitations of sample size, the infant study reported here provided a unique

opportunity to characterise a possible role of CMV infection on susceptibility to *M.tb* infection. We added some text to acknowledge the limitation of sample size in Line 627-637. Given our previous finding that CMV infection was associated with risk of TB, it is very relevant and important to look at CMV status in relation *M.tb* infection. We did not analyze immune correlates risk of *M.tb*-infection in CMV-positive and CMV-negative infants separately to tease out the impact of CMV on primary outcomes of this study, as the parent infant efficacy trial was not powered to detect differences between infants based on their CMV infection status. However, we were able to do this for the transcriptomic analysis and this is not limited by sample size because the algorithm in DESeq2 to find differentially expressed genes between *M.tb*-infected and *M.tb*-uninfected infants can borrow information from different genes, which can circumvent the issues associated with limited sample size.

Another major concern is the transient nature of the study outcome. Specifically, 14 of the 43 children who were identified as Mtb-infected based on QFT conversion from negative to positive during the study period, reverted to QFT negative by the end of study (Figure 2). Thus, for a large number of infants included in this analysis, they only transiently met the definition of Mtb-infection. The clinical significance of QFT reversion remains unknown as the authors acknowledge; however, there is either something immunologically distinct among the children who reverted or their initial QFT conversion was borderline initially and perhaps reflected a false-positive result. Either way, this is extremely problematic for an immune-based analysis where the sample size is too small for the reverters to be considered separately, and further confounds the results.

We have specifically selected infants who did not go on to develop TB disease and who did not receive chemoprophylaxis. The infants included in the study were QFT converters who likely controlled *M.tb* infection, hence the subsequent QFT reversion, though in this study we are comparing them with infants who were never *M.tb*-infected and thus our identification of correlates of that initial infection remain valid. QFT values for these reverters were well above the recommended cut-off (0.35 IU), 10/16 had QFT > 0.6 IU and 8/16 had QFT > 0.7 IU, so the initial QFT conversions were not borderline.

Regarding the significance of the question this study was designed to address—do immunologic risk factors for Mtb infection align with those identified for TB disease—to this reviewer, there is limited rationale to support this line of investigation. The findings of this study, although they may be of interest to those in the field of TB vaccine development, seem rather predictable.

Identification and validation of correlates of risk of *M.tb* infection and confirming or not whether there are common correlates of risk for both TB disease and *M.tb* infection is essential for the current strategy of using POI as a surrogate for POD to accelerate TB vaccine R&D. POI trials require smaller sample sizes and a shorter duration of follow-up compared to POD vaccine trials, and are therefore considered useful to provide a biological signal of efficacy prior to committing to POD trials. The success of this strategy relies on there being common correlates of risk of TB disease and *M.tb* infection. In this piece of work, we hypothesised that correlates of risk of TB disease would also be correlates associated with increased risk of *M.tb* infection. However, our findings show the converse – and this result was not predictable as this

reviewer suggests. In addition to investigating the primary parameters identified from the correlates of risk of disease analysis (Fletcher et al., 2016) and finding that they are not correlates of infection, our study provides new information on correlates of infection to advance the field; for example demonstrating an association between upregulation of immunoglobulin gene expression and increased risk of QFT conversion, which was not anticipated or predicted at the start of the study.

Several additional questions:

1) The Gene expression data analysis applies liberal definitions for DEGs, both with regards to the thresholds for a significant adjusted p-value (less than 0.2) and fold change (defining a fold change as 5% difference or more). What is the justification for these definitions? Related to this, *in Figure 5A, the legend states that q-values are shown, however in the actual figure it states p-value. This should be clarified.

1. The null hypothesis of differentially expression test in DESeq is $\text{LogFC} = 0$, which means that as long as the adjusted p value of the test is less than a certain value, they can be considered differentially expressed genes, no matter how large the LFC is.

2. FDR is the false positive rate, so if the threshold is set to 0.2, we can still expect that around 80% of DEGs identified are real DEGs. In statistics, the threshold for FDR can be set at greater than 0.05, such as 0.1 (the default setting for DESeq2), 0.2 or 0.3. There is precedent in other publications for using 0.2 as the threshold for FDR:

<https://doi.org/10.1172/jci.insight.132852>

<https://doi.org/10.1172/jci.insight.130090>

<https://doi.org/10.1186/s12916-019-1292-y>

4. The FDR threshold does not influence our pathway analysis based on the CERNO test, which is the result in Figure 3. Regarding Figure 5A, thank you for pointing out the mistake, it should be q value. We have corrected this in the revised manuscript.

2) Were HIV-exposed-uninfected infants included in this analysis? In the parent vaccine trial, infants had to be HIV ELISA negative, but this does not preclude HIV perinatal exposure as maternal HIV antibodies could be undetectable by age 4-6 months. Given concerns about abnormal immune function among HEU infants in the first year of life, as well as increased risk of Mtb-exposure among HEU infants, this should be considered. If maternal HIV testing was performed during the parent trial, these data could be used to address this consideration.

We did not test the mothers for HIV but did review the infants' clinic records, which indicate if they were exposed. Infants with mothers living with known HIV were excluded from the trial.

3) Is there a possibility that the plasma Ag85A-specific IgG responses detected on -D7 were maternal in origin? How was this excluded as a possibility? If this was not excluded, how do the authors interpret the findings?

It is possible that the 85A antibody results are confounded by maternal antibodies. However, even in the placebo group, we can see an increase of Ag85A-specific IgG on D28 compared to D-7 (among 72 infants in the placebo group, 72% of them had increased Ag85A-specific IgG levels in D28 compared to D-7, added to Supplementary figure 6). Therefore, we consider that most of the antibodies are not maternal but induced by BCG vaccination. We added discussion about this in Line 574-576.

4) The formatting for figure 3A is confusing and appears to contain unadjusted/uncorrected statistics. I am having difficulty understanding the two sets of p-values presented (y-axis versus overlaid).

P-values in the y-axis are the p-value of the difference between *M.tb* infected and uninfected infants (for the different immune parameters) using conditional logistic regression. P-values in the overlay shows that the p-values of the immune parameters measured by Multiplex assay are much lower than p-values of immune parameters measured by other assays. (The figure legend has now been edited to make this clearer).

5) The data presented in Figure 6 is difficult to understand given receipt of MVA vaccine by many study participants. What is the significant of these findings?

As stated in Lines 587-590 of the manuscript and in the comments made by Reviewer 2, the data presented in Figure 6 represents the association between the Ag85A specific IgG response and increased risk of *M.tb* infection, which might be partly attributed to the divergence between the BCG induced antibody response and T cell response, as shown by Figure 6. For the Ag85A IgG response at D-7, no infants had yet received MVA85A vaccination, so we analysed the combined data set of both groups of infants. As shown in the modified Figure 6 and Line 473-476, for the response at D28, we analysed the results for MVA85A and placebo separately, and the negative correlation between the T cell and antibody response only persisted in the placebo group.

6) Please review references, I noted several citations that appeared incorrectly placed.

We have checked all the references and adjusted where necessary and apologise for this error.

Reviewer #2 (Remarks to the Author):

The vast majority of individuals exposed to *Mtb* do not develop active tuberculosis disease, even if there is immunological evidence (by IGRA) of probable infection. Indeed, evidence from studies of healthcare workers in the preantibiotic era suggests that tuberculin skin test positivity, if not associated with progression to active TB, was actually protective against subsequent TB disease.

Furthermore, since they are easier and cheaper to run, there has been an interest in “prevention of infection” trials for candidate TB vaccines compared with “prevention of disease” trials. The assumption of these efficacy trials, is of course, that in an efficacious POI vaccine, those prevented from getting “infected” (assuming <100% efficacy) overlap with those that would develop TB disease. Investigating that assumption lies at the heart of this potentially important study from McShane and colleagues. The bottom line they present is that this may be an over-simplification.

The authors mine the rich clinical samples from the failed MVA85A candidate TB vaccine trial. Infants were all BCG vaccinated and then randomized to MVA85A or placebo. In the original study, ~70 infants developed active TB disease, split fairly evenly between both arms. In a post-hoc analysis, several features were identified to be potential correlates of protection against ATB: including HLA-DR upregulation on CD4+ T cells (risk of ATB), BCG-responsive ELISPOT and Ag85A-specific IgG responses _28 days post-MVA-A85A/placebo_.

A major strength of the study is that the infants defined as “infected” were rigorously evaluated as to not developing ATB over the course of the study in a naturalistic fashion. The authors examine the cohort of infants that were “infected” with Mtb (as defined by a single positive IGRA), but importantly, did not develop ATB during the course of the study AND did not receive isoniazid preventative therapy (IPT) [which would be expected to decrease risk of subsequent ATB in IGRA+ infants]. Thus these infants could be confidently assigned as not developing ATB in a naturalistic fashion during the course of the study. However, household contacts of ATB individuals were excluded from the analysis. It is not clear if this is because all HH contact infants received IPT [as guidelines suggest they should] or for some other reason. This is a shame, since if there were infants that were HH contacts and did not receive IPT and did not develop ATB, they would offer the strongest evidence of infection without ATB.

We completely agree that infants who were HH contacts and did not receive IPT would have made the best study group, but IPT is standard practice in South Africa, so it would be unethical and unacceptable to withhold IPT for those infants who were HH contacts.

A surprising observation from the study was that Ag85A-specific IgG responses (pre MVA85A/placebo) was associated with _increased_ risk of Mtb infection but not disease. Less surprisingly, general immune activation was associated with risk of Mtb infection. Ag85A-IgG responses were negatively correlated with BCG-specific ELISPOT, suggesting that some infants respond to BCG divergently in terms of humoral vs. cell-mediated responses.

Overall, this is a very important study. However, there are a few nuances in terms of interpretation of the data that should be addressed by the authors.

We thank the reviewer for this comment and for understanding the significance of this work. We agree that it is surprising that correlates of risk of disease were not associated with increased risk of infection; this highlights further the importance of conducting such studies. We also agree that the divergence between the antibody and T cell responses is important and should be investigated further in future.

Comments

1. A major point of discussion in the results is that in a former study (Fletcher et al 2016), Ag85A-specific IgG was anti-correlated with ATB and here it is correlated with Mtb infection but not ATB. But in that study, the Ab responses were defined _post-vaccination_ whereas here they are defined as pre-vaccination (i.e. in response to BCG). I’m not sure, but I don’t think the D-7 Ag85A responses were protective against

subsequent ATB – the authors should clarify. As such, the two studies were in reality studying different things: on the one hand, antibody responses to a subunit vaccine, and in the other, antibody responses to an attenuated live vaccine. As the other analyses show (i.e. B cell markers in the CMV-Mtb+ group), the risk of Mtb infection (but not disease) may be more reflective of the “skewed” response to BCG. This specific type of response (which is extremely unlikely to be limited to Ag85A) may or may not have been associated with protective against subsequent ATB, and requires clarification. As such, the wording e.g. lines 471-4 may need to be more nuanced than they currently are.

In both studies we looked at antibodies at both time points (D-7 and D28). In the disease cohort, although only D28 Ag85A-specific IgG were associated with a significantly reduced risk of TB disease, the D-7 Ag85A IgG also showed a trend towards an association with reduced risk of TB that did not reach statistical significance. In our current study we show an association between antibodies and increased risk of *M.tb* infection, but in this case, it was at D-7 not D28. We have modified the text (now Line 577-581) to clarify.

2. Somewhat related, in the _placebo_ arm, were D28 (i.e. after placebo) Ag85A-IgG titres positively correlated with subsequent “Mtb infection”?

We looked at the correlation between Ag85A-IgG on Day 28 and risk of infection by vaccination status, and Ag85A-IgG titres at D28 after placebo did not correlate with subsequent Mtb infection. We added this in Line 519-526 to make the analysis more comprehensive.

3. A limitation of the study is that “Mtb infection” is defined by IGRA positivity at a single time-point in the vast majority of the infants. It’s unclear to this reviewer what this means and whether it actually represents Mtb infection or instead Mtb exposure that did not establish infection (the field doesn’t really know). For example, the major “POI” trial to date is the BCG revaccination study (Nemes et al, NEJM, not cited). In that trial, BCG revaccination was not protective against a single positive IGRA, but it _was_ protective against “sustained” IGRA positivity. It can’t be helped that in this study, only two infants had sustained IGRA positivity, so the authors don’t have a choice other than to go with a single positive, but this limitation really needs to be explored more in the Discussion and discussed specifically in the context of the Nemes study. It may also warrant rewording of the title and abstract, replacing “Mtb infection” with “positive IGRA”

We agree with the reviewer that in retrospect, doing QFT at extra time points, in addition to the ones done in the study, would have been more informative. The QFT test was done 3 times during the study; at baseline, D336 and end of study (EOS). Infants who were selected for QFT conversion at D336 had another test at EOS, but only 2/16 remained positive. Although the QFT conversion is considered as a marker of infection, this QFT reversion could be considered as a protective immune response resulting in control of infection which in itself is interesting. As mentioned in the discussion (Line 562-565), in South African adolescents, TB incidence among QFT reverters was not different from those with sustained QFT reactivity, and they were both higher than TB incidence among sustained QFT negative adolescents.

Therefore, we did not think it is necessary to replace “Mtb infection” with “positive IGRA” in the title of this manuscript.

4. I am not familiar with South African TB guidelines, but generally, infants and children <5 years should be offered IPT as routine if they have a positive IGRA (or TST). If this is the case, the cases (“Mtb infection”) in this study may represent a biased/non-random sample of children within this prospective cohort. The authors should clarify the guidelines and discuss as appropriate.

South Africa, as a high burden country, does not routinely use the IGRA as a method of diagnosing *M.tb* infection, only TST is used. However, if an infant tested QFT positive within this trial, they were referred to the clinic for consideration of IPT – which was not always given because they might not be referred to TST test or they were TST negative. Similarly, we referred all infants with a history of household contact to the clinic for IPT. Once the referrals had been made, the decision as to whether to provide IPT or not was independent of the trial team.

5. Line 538. The authors state “we have previously shown that Ag85A-IgG was associated with reduced risk of disease” but they erroneously cite the Lu 2016 Cell paper (that did not show this) instead of the Fletcher 2016 study. This should be corrected.

We apologise for this error and have corrected this and checked all the other references as per our response to reviewer 1.

REVIEWER COMMENTS

Reviewer #1 (Remarks to the Author):

The authors' responses to the original critique are extensive and have improved the overall rigor and readability of the manuscript. I have a few remaining concerns:

Major: The discussion paragraph lines 553-563 are appreciated in highlighting the large number of QFT reverters in the current analysis and investigators' inability to compare immune phenotypes between sustained converters and reverters due to sample size limitations. However, the authors have not adequately discussed how this important limitation complicates interpretation of their findings. Specifically, in their analysis they are seeking to identify immunologic correlates associated with Mtb-infection. However, with the large number of QFT reverters they may have actually captured immunologic correlates of Mtb-clearance (eg, infants who successfully eliminated Mtb-infection) rather than correlates of Mtb-vulnerability. As this cannot be determined with the current data set, this limitation should be openly discussed.

Major: The investigators conclude that risk of Mtb-infection is associated with upregulation of several cytokines and chemokines reflective of immune activation. The concluding paragraph of the manuscript states "Our study identifies that inflammation and immune activation are associated with risk of M.tb infection in a subset of BCG-vaccinated South African infants who converted to positive QFT but did not progress to active TB disease." However, as the investigators clearly demonstrate, the impact of CMV infection on many of the key investigational findings (including cytokine profiles and cellular activation phenotypes) is profound. I am unclear if the data in the multiplex assay was analyzed in the context of CMV status (from Figure 3, and supplemental tables 3 & 5 it does not appear so)? If not adjusted for CMV, is there a risk that the observed results from the multiplex assay are being falsely interpreted? I do feel the transcriptomic data supports upregulation of genes associated with complement and immunoglobulin-genes among children who became Mtb-infected (regardless of CMV status), but am not as convinced that the investigators are seeing clear evidence of a more general inflammatory signature among these infants that is independent from CMV.

Minor: In the first paragraph of the discussion, would be helpful to restate the primary immune correlates of risk of TB disease that were re-evaluated here in context of Mtb-infection.

Minor: Discussion related to collection time points is appreciated (lines 504-526). However, this section could be shortened with consideration for moving the data presented in lines 520-523 into the corresponding results section. Lines 515-517, specifically, should be edited for clarity.

Minor: Regarding the data presented in Supplemental Table 7B, it appears that there is a column missing in the section "additional logistic regression;" there are long lists of p-values and OR listed but no key to denote what variables they relate to.

Reviewer #2 (Remarks to the Author):

Overall, the authors have responded to my comments. In the response with regards to why the IGRA+ infants did NOT receive IPT, they comment that the infants were referred for IPT, but did not receive it, _possibly because they were TST negative_ (which is the threshold for IPT in South Africa). Are the TST results of these infants known to the manuscript authors? If so, this should be included in the supplementary information (or stated in the Methods that this is the reason for the lack of IPT and the TST result is not known). Although I'm not sure what a discordant TST/IGRA means in this specific context if I'm honest, future studies looking at the discrepancy between infection/ disease in clinical trials, may also record these data and it would be important to document. I also think that the authors should at least mention in the Discussion that these infants who didn't receive IPT may be somewhat unrepresentative of all IGRA+ infants.

REVIEWER COMMENTS

We thank the reviewers for their careful evaluation of our resubmission and helpful comments which we believe have further improved the manuscript. We have the following additional responses:

Reviewer #1 (Remarks to the Author):

The authors' responses to the original critique are extensive and have improved the overall rigor and readability of the manuscript. I have a few remaining concerns:

Major: The discussion paragraph lines 553-563 are appreciated in highlighting the large number of QFT reverters in the current analysis and investigators' inability to compare immune phenotypes between sustained converters and reverters due to sample size limitations. However, the authors have not adequately discussed how this important limitation complicates interpretation of their findings. Specifically, in their analysis they are seeking to identify immunologic correlates associated with Mtb-infection. However, with the large number of QFT reverters they may have actually captured immunologic correlates of Mtb-clearance (eg, infants who successfully eliminated Mtb-infection) rather than correlates of Mtb-vulnerability. As this cannot be determined with the current data set, this limitation should be openly discussed.

We thank the reviewer for this helpful comment and have modified the discussion as follows (Line 585-593):

As samples from infants who went on to develop TB disease were used in our previous correlates of risk of disease analysis, in this study we have evaluated immune correlates of risk of *M.tb* infection for a subset of infants who did not go on to develop TB disease. Thus these infants could be considered to be protected from TB disease, in parallel with being susceptible to *M.tb* infection. This is particularly true as the majority of infants infected with *M.tb* at Day 336 of the study (14/16) reverted to negative QFT reactivity during the study follow-up period. The immune parameters we have identified here may therefore associate with successful clearance of *M.tb* infection. Further work on this is needed to gain further insights into protective immunity against *M.tb* infection.

Major: The investigators conclude that risk of Mtb-infection is associated with upregulation of several cytokines and chemokines reflective of immune activation. The concluding paragraph of the manuscript states "Our study identifies that inflammation and immune activation are associated with risk of M.tb infection in a subset of BCG-vaccinated South African infants who converted to positive QFT but did not progress to active TB disease." However, as the investigators clearly demonstrate, the impact of CMV infection on many of the key investigational findings (including cytokine profiles and cellular activation phenotypes) is profound. I am unclear if the data in the multiplex assay was analyzed in the context of CMV status (from Figure 3, and supplemental tables 3 & 5 it does not appear so)? If not adjusted for CMV, is there a risk that the observed results from the multiplex assay are being falsely interpreted?

The analysis of data from the multiplex assay by *M.tb* infection status was not adjusted for CMV infection status. However, when we looked separately at the influence of CMV infection status on the data from the multiplex assay, only 3 of 45 analytes were significantly associated with CMV infection (Benjamini-Hochberg adjusted p-value < 0.2, adjusted only within the multiplex assays), compared with 26 out of 45 analytes that were significantly associated with risk of subsequent *M.tb* infection. Although three multiplex analytes (IP-10, TNF- α and Complement C2) were significantly associated with both CMV infection and risk of *M.tb* infection, the association between them and risk of *M.tb* infection may still be valid since they may contribute to the association between CMV infection and risk of *M.tb* infection. Line 405-417 and 526-531 were modified to discuss this in detail as follows:

To ensure that the increase of immune parameters measured by multiplex assays in *M.tb*-infected infants compared to *M.tb*-uninfected infants was not confounded by the CMV infection status of the infants, we evaluated the association between these immune parameters and CMV infection and found that IP-10, TNF- and complement C2 were all significantly upregulated in CMV-infected infants compared with CMV-uninfected infants (FDR \leq 0.2, adjusted only within immune parameters measured by multiplex assays; Supplementary Table 7B). The plasma levels of these three proteins were also significantly upregulated in infants who became infected with *M.tb* (FDR \leq 0.2; Supplementary Table 5). Only 3 out of the 45 immune parameters were significantly associated with CMV infection, compared to 26 out of the 45 immune parameters that were significantly associated with risk of subsequent *M.tb*-infection, suggesting that the association between most of the immune parameters measured by multiplex assays and risk of *M.tb*-infection was not confounded by CMV infection.

As we show in the results, only 3 out of these 45 immune parameters were associated with CMV infection of the infants, suggesting that the association between most of these immune parameters and risk of *M.tb*-infection were not confounded by CMV infection. The 3 immune parameters associated with both CMV infection and risk of *M.tb*-infection, IP-10, TNF- α and Complement C2, might contribute mechanistically to the increased risk of *M.tb*-infection in CMV-infected infants.

I do feel the transcriptomic data supports upregulation of genes associated with complement and immunoglobulin-genes among children who became Mtb-infected (regardless of CMV status), but am not as convinced that the investigators are seeing clear evidence of a more general inflammatory signature among these infants that is independent from CMV.

We thank the reviewer for this comment. As the reviewer pointed out, we did see an upregulation of gene sets related to the inflammatory response, such as type I interferon response, viral sensing & immunity in *M.tb*-infected compared to *M.tb*-uninfected infants when we analysed all infants or CMV-infected infants, but not for CMV-uninfected infants. We acknowledge that the inflammatory signature upregulated in *M.tb*-infected infants is dependent on CMV and we added this point into discussion (Line 662-667) and as below:

In our transcriptomic datasets, we found gene sets related to the inflammatory response, such as type I interferon response, viral sensing & immunity, were upregulated in *M.tb*-infected compared to *M.tb*-

uninfected infants when we analysed all infants combined or CMV-infected infants alone, but not CMV-uninfected infants. This suggests the upregulation of inflammatory signatures we found in *M.tb*-infected compared to *M.tb*-uninfected infants for all infants combined was mainly driven by CMV infection.

Minor: In the first paragraph of the discussion, would be helpful to restate the primary immune correlates of risk of TB disease that were re-evaluated here in context of Mtb-infection.

We thank the reviewer for this suggestion. We have now restated the primary immune correlates of risk of TB disease in the first paragraph of the discussion as follows:

In this study we primarily investigated whether the three immune correlates of risk of TB disease (HLA-DR expression on CD4⁺ T-cells, frequencies of BCG-specific IFN- in PBMC and plasma Ag85A-specific IgG responses on Day -7) were also associated with susceptibility to *M.tb* infection in BCG-vaccinated South African infants, using the same protocols to ensure comparability of the results obtained from both studies [6].

Minor: Discussion related to collection time points is appreciated (lines 504-526). However, this section could be shortened with consideration for moving the data presented in lines 520-523 into the corresponding results section. Lines 515-517, specifically, should be edited for clarity.

We thank the reviewer for the suggestions. We have moved Line 520-523 (now Line 368-374) to the results and modified the text from Line 504-526 (now Line 539-557) and Line 342-347 to make it clearer.

Minor: Regarding the data presented in Supplemental Table 7B, it appears that there is a column missing in the section “additional logistic regression;” there are long lists of p-values and OR listed but no key to denote what variables they relate to.

Since there is no Supplementary Table 7B in the previous version of this manuscript, we believe this comment pertains to Supplementary Table 6B. For this table, there are two types of odds ratio and p-value. One is the odds ratio and p-value when we use only Day -7 samples for conditional logistic regression, the other is the odds ratio and p-value when we use both Day -7 and Day 28 samples combined for conditional logistic regression. Therefore, we do not think there is any column missing. We have added this into the Table legend to make it clearer.

Reviewer #2 (Remarks to the Author):

Overall, the authors have responded to my comments. In the response with regards to why the IGRA+ infants did NOT receive IPT, they comment that the infants were referred for IPT, but did not receive it, _possibly because they were TST negative_ (which is the threshold for IPT in South Africa). Are the TST results of these infants known to the manuscript authors? If so, this should be included in the supplementary information (or stated in the Methods that this is the reason for the lack of IPT and the TST result is not known). Although I'm not sure what a discordant TST/IGRA means in this specific context if I'm honest, future studies looking at the discrepancy between infection/ disease in clinical trials, may also

record these data and it would be important to document. I also think that the authors should at least mention in the Discussion that these infants who didn't receive IPT may be somewhat unrepresentative of all IGRA+ infants.

We thank the reviewer for the suggestions. We have added Line 682-685 to discuss this potential bias:

In this study, we excluded infants who received IPT, which might include infants who were QFT-positive and did not progress to TB disease but received IPT because they were also TST-positive. For the infants included in our study, we did not know their TST status; it is therefore possible that these infants were not representative of all QFT-positive infants.